# Skip-It? Theoretical Conditions for Layer Skipping in Vision–Language Models

**Max Hartman** [* 1]  **Vidhata Jayaraman** [* 1 2]  **Moulik Choraria** [1]  **Akhil Bhimaraju** [1]  **Lav Varshney** [3]

## Abstract

Vision–language models achieve incredible performance across a wide range of tasks, but their large size makes inference costly. Recent work has shown that multimodal processing contains significant redundancies, making it possible to skip certain layers with minimal performance loss. Yet current pruning techniques remain ad-hoc, relying on heuristics or hyperparameter sweeps rather than principled criteria for determining when layer skipping is beneficial. In this paper, we propose a unified framework that characterizes the redundancy conditions under which pruning can enhance efficiency without sacrificing performance. Central to our approach are experimentally verifiable and interpretable notions of redundancy that can be evaluated without requiring downstream task performance as a metric. Applying this framework, we corroborate prior findings that both early and late vision tokens are redundant across models, and we validate our conditions by showing they align with actual performance degradation. Beyond these empirical results, our framework provides a theoretically grounded understanding of redundancy in VLMs and unifies many of the ideas behind modern layer-skipping techniques.

## 1. Introduction

Vision-language models (VLMs) such as BLIP (Li et al., 2022), LLaVA (Liu et al., 2023; 2024a), and more recently, Qwen (Bai et al., 2023; Qwen et al., 2025), Deepseek-VL (Lu et al., 2024), and Gemma (Gemma Team et al., 2025)

have become popular due to their impressive performance on a wide range of vision-language tasks. Nevertheless, their escalating training and inference costs (Jin et al., 2025) creates barriers to widespread adoption. This computational burden stems from how these models typically build upon large language model (LLM) backbones, requiring the processing of extensive sequences of vision tokens alongside text, particularly when handling high-resolution images.

Consequently, techniques to improve VLM efficiency while maintaining performance have become an active area of research. Many approaches build upon efficiency methods from large language models (LLMs) and can be broadly classified as training-time or inference-time improvements. Training time methods include parameter-efficient approaches such as LoRA (Hu et al., 2022; Dettmers et al., 2023; Biderman et al., 2024) and and architectural modifications like Mixture of Experts (MoE) (Artetxe et al., 2022; Bao et al., 2022; Lin et al., 2026). Two common Inference time methods include token compression and layer skipping which will be the main topics explored in this work. Token compression techniques (Chen et al., 2024b; Vasu et al., 2025; Liu et al., 2025) reduce redundancy in vision representations, while layer skipping, initially developed for efficient LLM inference (Elhoushi et al., 2024), bypasses unnecessary computation during forward passes. Recent VLM adaptations of the latter include AdaSkip (He et al., 2025), FlexiDepth (Luo et al., 2025a), and retrained models like DeepInsert (Choraria et al., 2026), MoLe-VLA (Zhang et al., 2026), and $\gamma$-MoD (Luo et al., 2025b). Despite their empirical success, these methods lack a principled basis for determining which layers to skip.

Our contributions are as follows:

1. We propose a learning- and information- theoretic framework for characterizing redundancy in VLMs. Our framework provides easy-to-compute conditions for when pruning can enhance efficiency without performance degradation and unifies several inference-time layer skipping techniques.

2. We use our theory to empirically determine where redundancy exists in VLMs. We find large redundancy in the early and late vision tokens across models which validates existing pruning methods.

---

[*]Equal contribution [1]Department of Electrical and Computer Engineering, The University of Illinois, Champaign, IL, United States [2]Department of Mathematics, The University of Illinois, Champaign, IL, United States [3]AI Innovation Institute, Stony Brook University, Stony Brook, NY, United States. Correspondence to: Max Hartman <maxh3@illinois.edu>, Vidhata Jayaraman <vidhata2@illinois.edu>.

*Proceedings of the 43rd International Conference on Machine Learning*, Seoul, South Korea. PMLR 306, 2026. Copyright 2026 by the author(s).

3. We then demonstrate that our redundancy conditions can predict when layer skipping can be applied with minimal performance degradation. To make this framework operational, we introduce a label-free algorithm that applies vision-token layer skipping with a small subset of data, bridging the gap between theoretical redundancy and practical deployment.

By providing rigorous, measurable notions of redundancy, we hope to inspire future work on token and layer reduction methods.

## 2. Related Work

### 2.1. Vision Language Models

VLMs feed vision and textual tokens into an autoregressive LLM. BLIP models (Li et al., 2022; 2023) were some of the first to introduce vision-language pretraining, with the latter using a Q-former to connect frozen image encoders with LLMs. Flamingo introduced the ability to ingest mixed insertions of images, videos, and text (Alayrac et al., 2022). LLaVA introduced a multimodal instruction-following model. LLaVA-NeXT built on LLaVA with improved reasoning and performance (Liu et al., 2024a). More recently, Molmo improved capabilities in pointing and counting tasks (Deitke et al., 2025). Llama-4 introduced a mixture-of-experts VLM with a massive context length. Specifically, Llama-4 Scout has a token context length of 10M (Meta AI, 2024). In most models, the vision tokens are obtained from a pretrained vision encoder, such as CLIP (Radford et al., 2021) or SigLIP (Zhai et al., 2023). Some research questions about these models include inference-time inefficiency, multimodal alignment, and vision interpretability.

### 2.2. Layer Skipping and Pruning

Redundancies in tokens and layers have motivated efficiency techniques in both LLMs and VLMs. In LLMs, Shukor & Cord (2024) introduced a method for skipping certain computations (whether entire blocks, specific feed-forward networks or self-attention layers). AdaSkip introduces sublayer-wise skipping for long-context inference (He et al., 2025), while FlexiDepth enables adaptive layer skipping (Luo et al., 2025a). Both exploit redundancy to reduce computation while maintaining performance. Multimodal models have adopted similar strategies: $\gamma$-MoD converts dense layers into sparse Mixture-of-Depth layers (Luo et al., 2025b), MoLe-VLA applies layer skipping to robot manipulation (Zhang et al., 2026), Skip-Vision (Zeng et al., 2025) skips certain feed-forward networks during training and prunes certain key-value pairs during inference, and DeepInsert injects vision tokens into later layers to reduce early-layer overhead (Choraria et al., 2026). Parallel work focuses on token reduction through multimodal skipping or early exit. FastV prunes visual tokens by learning attention patterns (Chen et al., 2024b), Visual Token Withdrawal (VTW) removes visual tokens after certain layers (Lin et al., 2025), PruMerge leverages visual encoder sparsity to discard tokens (Shang et al., 2025), and $ST^3$ (Zhuang et al., 2025) prunes redundant vision tokens across layers and dynamically reduces the number of vision tokens across layers (Zhuang et al., 2025). Our framework specifically aims to unify these multimodal layer skipping techniques but can possibly be extended to unify inference-time pruning techniques in general (see Section 5).

## 3. Framework for Measuring Redundancy

### 3.1. Definitions of Redundancy

Intuitively, a neural network component is a candidate for pruning when it becomes redundant. A layer is redundant if it acts as an identity function, whereas tokens are redundant if their representations encode duplicate information. In practice, this redundancy is often detected by measuring the cosine distance between activations.

To provide a unified framework for analyzing both layer and token pruning, we abstract these activations as random variables (RVs). With this formalization, we can rigorously capture the redundancy between two components using the expected cosine distance between their corresponding RVs. We refer to this metric as geometric redundancy.

*Definition* 1 (Geometric redundancy). Let $\rho : \mathcal{X} \times \mathcal{X} \to [0, \infty)$ be a symmetric function (e.g. cosine distance or a metric). Then given random variables $X_{\ell-1}, X_\ell$ and $\epsilon > 0$, geometric $\epsilon$-redundancy (or geometric redundancy) is $\mathbb{E}[\rho(X_{\ell-1}, X_\ell)] < \epsilon$.

We similarly define proximal redundancy as being when two RVs are close together with high probability.

*Definition* 2 (Proximal Redundancy). Let $\rho : \mathcal{X} \times \mathcal{X} \to [0, \infty)$ be a symmetric function (e.g. cosine distance or a metric), $t$ be some threshold value, and $\epsilon > 0$. Then random variables $X_{\ell-1}, X_\ell$ are $t$-proximal with probability $1 - \epsilon$ (or proximally redundant) if $\mathbb{P}[\rho(X_\ell, X_{\ell-1}) < t] \geq 1 - \epsilon$.

These measures capture how much the layer/token representation has changed. However, a small change in representation does not guarantee a small change in downstream performance or a large statistical connection between the two RVs. To address this, we introduce operational notions of redundancy that measure how much the output is affected. Specifically, we define functional redundancy (does downstream performance change?) and informational redundancy (is new information added?).

*Definition* 3 (Functional redundancy). Given a task variable $Z$, two random variables $X_{\ell-1}, X_\ell$, and $\epsilon > 0$, functional

$\epsilon$-redundancy (functional redundancy) is $\mathbb{E}[\|\mathbb{E}[Z|X_\ell] - \mathbb{E}[Z|X_{\ell-1}]\|_2^2] < \epsilon$.

*Definition* 4 (Informational redundancy). Given random variables $X_{\ell-1}, X_\ell$ and $\epsilon > 0$, informational $\epsilon$-redundancy (informational redundancy) is $H(X_\ell|X_{\ell-1}) < \epsilon$ where $H(\cdot|\cdot)$ is conditional entropy.

In this framework, functional redundancy ensures that the layer/token's contribution is negligible for the specific task $Z$ (e.g., the classification logits remain stable). Informational redundancy requires that $X_\ell$ is almost deterministic given $X_{\ell-1}$, meaning the layer/token adds no new information. The remainder of this section connects these four notions. Specifically, we show that geometric and proximal redundancy—while less interpretable—imply these more operational forms under natural assumptions.

### 3.2. Functional Redundancy

We begin by analyzing functional redundancy, which measures the difference between optimal estimators on a task $Z$.

*Theorem* 1. Let $X_\ell, X_{\ell-1}$ be unit-norm random variables and $Z$ be the random variable of predictive interest (e.g. normalized hidden representations of layers $\ell, \ell - 1$ and the task ground truth respectively). Let $\rho(x,y) = 1 - \frac{\langle x,y \rangle}{\|x\|\|y\|}$. Assume $\mathbb{E}[\rho(X,Y)] < \frac{\epsilon}{2}$ and that

$$h(x,y) = E[Z|X_\ell = x, X_{\ell-1} = y]$$

is $\alpha$-Lipschitz in the first argument and $\beta$-Lipschitz in the second. Then $E[\|\|E[Z|X_\ell] - E[Z|X_{\ell-1}]\|_2^2] < 2(\alpha^2 + \beta^2)\epsilon$.

*Proof.* For the proof, see Appendix A.2. □

Theorem 1 establishes that under some regularity assumptions, the commonly used geometric redundancy can imply functional redundancy. In other words, a minimal average cosine distance between layer/token representations can guarantee similar performance on a downstream task. However, in practice, we can never achieve these optimal estimators, so Theorem 2 bounds empirical estimates of these estimators.

*Theorem* 2. Let $X_\ell, X_{\ell-1}$ be unit-norm random variables and $Z$ be the random variable of predictive interest, as before. Let $\rho(x,y) = 1 - \frac{\langle x,y \rangle}{\|x\|\|y\|}$. Assume $\mathbb{E}[\rho(X,Y)] < \frac{\epsilon}{2}$ and that

$$h(x,y) = E[Z|X_\ell = x, X_{\ell-1} = y]$$

is $\alpha$-Lipschitz in the first argument and $\beta$-Lipschitz in the second. Let $\hat{f}_\ell$ be a finite-sample estimate of $f_\ell^*(x) = \mathbb{E}[Z|X_\ell = x]$ and $\hat{f}_{\ell-1}(x)$ be a finite-sample estimate of $f_{\ell-1}^*(x) = \mathbb{E}[Z|X_{\ell-1} = x]$. Further let, $\eta_\ell = \mathbb{E}[\|\hat{f}_\ell(X_\ell) -$ $f_\ell^*(X_\ell)\|^2]$ and $\eta_{\ell-1} = \mathbb{E}[\|\hat{f}_{\ell-1}(X_{\ell-1}) - f_{\ell-1}^*(X_{\ell-1})\|^2]$. Then:

$$\mathbb{E}[\|\hat{f}_\ell(X_\ell) - \hat{f}_{\ell-1}(X_{\ell-1})\|^2] < 3\eta_\ell + 3\eta_{\ell-1} + 6(\alpha^2 + \beta^2)\epsilon.$$

*Proof.* We prove this by again converting from cosine distance to mean squared error and then rewriting $\hat{f}_\ell(X_\ell) - \hat{f}_{\ell-1}(X_{\ell-1})$ as $(\hat{f}_\ell(X_\ell) - f_\ell^*(X_\ell)) + (f_{\ell-1}^*(X_{\ell-1}) - \hat{f}_{\ell-1}(X_{\ell-1})) + (f_\ell^*(X_\ell) - f_{\ell-1}^*(X_{\ell-1}))$. We then upper bound $\hat{f}_\ell(X_\ell) - \hat{f}_{\ell-1}(X_{\ell-1})$ using this new form by invoking Theorem 1 along with the definitions of $\eta_\ell$ and $\eta_{\ell-1}$. For more details, refer to Appendix A.2 □

Thus, under the same regularity assumptions, we obtain guarantees for empirical estimators that mirror those for optimal ones. Importantly, the bound still decays linearly in $\epsilon$, showing that empirical functional redundancy inherits the same behavior. Previous layer skipping work has employed similar Lipschitz assumptions on self-attention and feed-forward networks, notably Zeng et al. (2025). Refer to Appendix A.4 for a brief study on the Lipschitz constants observed in practice.

### 3.3. Informational Redundancy

We now turn to informational redundancy, which asks whether one RV can be (nearly) determined from a second RV. This is formalized via the conditional entropy $H(X_\ell|X_{\ell-1})$. Specifically, we show that if two random variables $X_\ell$ and $X_{\ell-1}$ are proximally redundant, then they are also informationally redundant. In the context of layer skipping, this redundancy notion is not very meaningful because the current layer is a deterministic function of the previous. However, for token pruning, the predictability of one token given another can still provide a useful interpretation of the redundancy. These results rely heavily on the distance-based Fano's inequality (Duchi & Wainwright, 2013) and its generalizations (Braun & Pokutta, 2015).

We first get an upper bound on $H(X_\ell|X_{\ell-1})$ from Braun & Pokutta (2015). If this upper bound is sufficiently low, this shows a high statistical correlation between $X_\ell$ and $X_{\ell-1}$. We then use a complementary lower bound from Braun & Pokutta (2015) and Duchi & Wainwright (2013) on the mutual information $I(X_\ell; X_{\ell-1})$. If this lower bound is sufficiently large, we can then show that there is a large amount of information shared between the two RVs, thereby also implying a large amount of redundancy. For more details on the specifics of these bounds, refer to Theorems 3, 4, and Corollary 4 in Appendix A.2.

There is substantial prior work, such as (Xu et al., 2020; Dissanayake et al., 2025), which has shown that concepts like usable information and unique/redundant information are useful for analyzing machine learning paradigms. See

Appendix A.3 for further discussion of the connection to partial information decomposition (PID).

## 3.4. Relating all notions of redundancy

We now give some final results to connect all notions of redundancy.

*Proposition* 1. Let $\rho : \mathcal{X} \times \mathcal{X} \to \mathbb{R}$ be a symmetric function with $0 \leq \rho \leq 1$. Then

$$\mathbb{P}[\rho(X, Y) > t] < \frac{\epsilon - t}{1 - t} \text{ implies } \mathbb{E}[\rho(X, Y)] < \epsilon,$$

*Proof.* Apply the tail integration formula and split the integral into a part from 0 to $t$ and another part from $t$ to 1. Refer to Appendix A.2 for more details. □

Thus, under natural assumptions, proximal redundancy implies geometric redundancy. In other words, if two RVs (e.g., layers/tokens) are close together with high probability, then they are close together on average.

Theorem 5 shows that under some additional natural Markov and boundedness assumptions, informational redundancy implies functional redundancy. In particular, since the hidden states are taken after adding the residual, this Markov assumptions holds.

*Theorem* 5. Suppose there are random variables $Z, X_\ell, X_{\ell-1}$ with $Z \in \mathbb{R}^d$ and $X_\ell, X_{\ell-1}$ being continuous unit-norm random variables. Further suppose $\|Z\|_2 \leq B$ almost surely and that $X_{\ell-1} \!-\! X_\ell \!-\! Z$ is a Markov chain. Then $\mathbb{E}[\|\mathbb{E}[Z|X_\ell] - \mathbb{E}[Z|X_{\ell-1}]\|_2^2] \leq 2B^2 I(Z; X_\ell | X_{\ell-1})$. If, in addition, there exists finite $C$ such that $H(X_\ell | Z, X_{\ell-1}) \geq -C$ then $\mathbb{E}[\|\mathbb{E}[Z|X_\ell] - \mathbb{E}[Z|X_{\ell-1}]\|_2^2] \leq 2B^2(H(X_\ell|X_{\ell-1}) + C)$. In particular, if $X_\ell$ is discrete then $C = 0$ and if $p_{X_\ell | Z, X_{\ell-1}}(x) \leq M$ for all $x$ then $C = \log M$.

*Proof.* We prove this first by expressing $\mathbb{E}[Z|X_\ell = a] - \mathbb{E}[Z|X_{\ell-1} = b] = \int_{\mathbb{R}^d} z(p_{Z|X_\ell=a}(z) - p_{Z|X_{\ell-1}=b}(z))dz$. We can then bound $\|\mathbb{E}[Z|X_\ell = a] - \mathbb{E}[Z|X_{\ell-1} = b]\|$ using the triangle inequality. We then use Pinsker's inequality to bound $\int_{\mathbb{R}^d} |p_{Z|X_\ell=a}(z) - p_{Z|X_{\ell-1}=b}(z)|dz$ using KL divergence. Since $X_{\ell-1} \!-\! X_\ell \!-\! Z$ is a Markov chain, we can convert this into an upper bound using conditional entropy (Lemma 4). For more details, refer to Appendix A.2. □

Since differential entropy can be negative, one cannot get a direct bound between functional redundancy and informational redundancy in general. But, by considering RVs to be discrete (say, due to quantized activations after compression or finite computer memory), then informational redundancy directly implies functional redundancy. Thus, informational redundancy can be viewed as a more fundamental type of redundancy than functional redundancy. Intuitively, if one

layer/token representation is very predictable from another, then their optimal performance on a downstream task will be comparable. Furthermore, the connection to PID gives an interesting way to interpret this bound: under certain conditions, the average difference in performance between optimal MSE estimators based on random variables $X$ and $Y$ is upper bounded by unique information about $X$ that only $X$ has. More details on PID are in Appendix A.3.

### Key Insight #1

Our theory formally establishes that empirical notions of redundancy imply more meaningful notions of redundancy. Specifically, closeness of layer/token representations ensures functional and information theoretic guarantees on downstream usage.

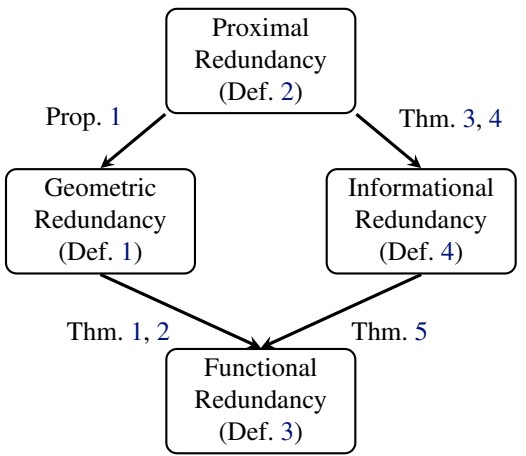

*Figure 1.* Implication relationships among different notions of redundancy.

## 4. Unification of Layer Skipping

In this section, we generalize layer skipping methods into Early Exit and Late Entry, arguing that a number of existing layer skipping methods are subsumed by this formulation. We experimentally find when layer skipping is viable using our redundancy framework, and then validate these predictions by showing that when the conditions are met, model performance degrades minimally (and vice versa). Lastly, we provide a generalization bound and experimental validation to show that in practice, a small subset of samples from the dataset is enough to determine the viability of skipping, without access to ground truth labels.

### 4.1. Defining Layer Skipping

To formally define layer skipping, consider a VLM with $n$ layers, $\phi_\theta^n$. Let $X = (X^{(text)}, X^{(vis)})$ be the input to the VLM. We consider late entry, say of the vision tokens, in the first $\ell$ layers to be $\phi_\theta^{n-\ell}((\phi_\theta^\ell(X^{(text)}), X^{(vis)}))$, and

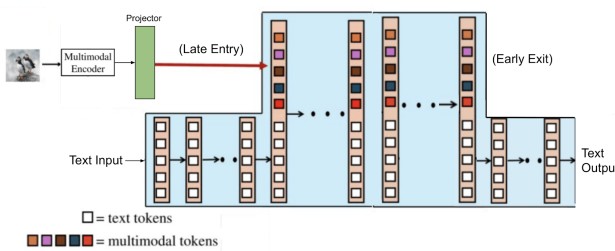

*Figure 2.* Early Exit and Late Entry. Specifically, the visual tokens are not passed into the first few layers but instead, directly inserted along with the prompt to the chosen layer for insertion or the vision tokens are removed from the forward pass after a certain layer.

early exit to be $\left(\phi_\theta^\ell\left(\left[\phi_\theta^{n-\ell}(X)\right]^{(text)}\right),\left[\phi_\theta^{n-\ell}(X)\right]^{(vis)}\right)$, where the vision tokens skip the last $\ell$ layers. A visual depiction can be seen in Figure 2.

We motivate the possibility of late entry layer skipping through the following intuition: given an insertion layer $L$, if the vision activations $V_l$, $l < L$, undergo minimal change (under some suitable metric) and the effect of vision tokens on text representations (via attention) is minimal, then the output with late entry of vision will be close to the true original output at layer $L$. In addition to this result, we also have that if the vision activations in the final layers undergo minimal change and the effect of vision tokens on text representations is minimal in the final layers, then one can perform early exit. We formally state and prove these as Theorem 6 and Lemma 5 in Appendix A.2.

Note that redundancy is clearly a necessary condition for late entry. If the visual representations have evolved too much, then simple insertion of the original representations at layer $L$ may not necessarily produce quality outputs. Since in the remaining layers, visual tokens can still attend to textual tokens, having minimal cross attention does not clearly indicate a condition to do late entry layer skipping.

However, for early exit, minimal cross attention from vision to text is sufficient because the output modality is text. If the vision tokens are minimally attending to the text, these tokens can be skipped because they are minimally changing the output tokens. Therefore, the vision tokens can be skipped with minimal output degradation. This is quantified using the visual attention ratio (VAR) (Jiang et al., 2025).

Many existing techniques can be characterized as either Early Exit or Late Entry methods. Early Exit includes (but is not limited to) VTW (Lin et al., 2025), which uses a a KL divergence criterion to determine when to exit vision tokens, and LayerSkip (Elhoushi et al., 2024), which proposes an LLM training method allowing for early exit. Late Entry includes DeepInsert (Choraria et al., 2026), which withholds vision tokens until a set layer, and similarity-aware token

pruning (SAINT) (Jeddi et al., 2025), which leverages token similarities to prune tokens from early layers.

One commonality across all of them is some heuristic measure of "redundancy" to determine which layers are viable for skipping. However, what is missing is: 1) a principled justification for how this measure equates to operational redundancy and 2) a systematic way to exploit this redundancy, that crucially, can generalize across different models and datasets without needing ground truth labels. The framework proposed here provides a rigorous justification for using a heuristic (such as cosine distance) for redundancy and then further justifies the use of layer skipping from this redundancy.

### 4.2. Empirical Viability of Early Exit and Late Entry Layer Skipping

As described in the previous section, for late entry techniques to be viable, we need geometric and proximal redundancy, and for early exit techniques to be viable, minimal cross attention from vision to text is sufficient. In this section, we determine when these conditions are met empirically, across different models, datasets, and tasks.

#### 4.2.1. EXPERIMENTAL SETUP

We consider the hidden states (extracted post residual addition) of each token across each layer and compute both the average cosine distance and the probability of a small average cosine distance between adjacent hidden states. We separate vision and textual tokens so that we can evaluate their differences. Formally, let $H_T$ and $H_V$ denote the sets of textual and visual hidden states for all layers. For token $i$ at layer $\ell$, let $h_{\ell,i}$ denote its hidden state. We define the average cosine distance between adjacent layers as

$$\mathcal{D}_\ell^{(M)} := \frac{1}{N_M}\sum_{i=1}^{N_M}\rho\big(h_{\ell,i}^M, h_{\ell-1,i}^M\big), \qquad (1)$$

and the probability of adjacent hidden states being close as

$$p_\ell(t; H_M) := \frac{1}{N_M}\sum_{i=1}^{N_M}\mathbb{1}\{\rho\big(h_{\ell,i}^M, h_{\ell-1,i}^M\big) < t\}, \quad (2)$$

where $M \in \{V, T\}$ refers to the modality (vision or text), $t$ is a user-specified threshold, $N_T = |H_T^{(\ell)}|$ and $N_V = |H_V^{(\ell)}|$ are the number of textual and visual tokens, respectively, $h_{\ell,i}^T, h_{\ell-1,i}^T \in H_T$, $h_{\ell,i}^V, h_{\ell-1,i}^V \in H_V$, and $\rho(\cdot, \cdot)$ denotes cosine distance. This is done for $\ell \in [1, N]$, where $N$ is the number of layers in the specific model.

#### 4.2.2. MODELS & DATASETS

For our experiments, we use LLaVA 1.5 (7B and 13B) (Liu et al., 2023) and LLaVA NeXT (1.6) (Liu et al., 2024a) as

VLMs. Additional results using DeepSeek-VL (7B base) (Lu et al., 2024) and Qwen 2.5 VL (Qwen et al., 2025) are in Appendix C.

For our experiments, we consider both multiple choice and free response datasets, spanning a diverse set of vision-language tasks including general question answering (GQA (Hudson & Manning, 2019), VQA (Antol et al., 2015), Visual7W (Zhu et al., 2016)), text, OCR, and document-based (AI2D (Kembhavi et al., 2016), OCRBench (Liu et al., 2024b), TextVQA (Singh et al., 2019)), and multimodal reasoning (MMMU (Yue et al., 2024), RealWorldQA (xAI, 2024), MMStar (Chen et al., 2024a), MathVision (Wang et al., 2024)). Further details on dataset and evaluation protocols are in Appendix B.

#### 4.2.3. GEOMETRIC AND PROXIMAL REDUNDANCY ANALYSIS

Figure 3 shows that the early layer vision tokens have very low cosine distances with each other and very high probability of being below the cosine distance threshold (for $t = 0.01, 0.025, 0.05$). In the later layers of LLaVA 1.5 7B and 13B, both the early and textual tokens demonstrate proximal redundancy. In LLaVA 1.6, this trend is visible but not as extreme. Note that the threshold values are arbitrary and are varied to show stability; we mainly seek to highlight that early vision tokens have low high probabilities of being close, which means that there are redundancies via Theorems 1—4.

Please refer to Appendix C for additional results. We verify that this is trend is consistent when using other distance metrics, such as centralized kernel alignment (CKA) (Kornblith et al., 2019) in Appendix C. Our results indicate that these models exhibit clear redundancy that can be exploited for efficiency improvements. Next, we validate that the inter-modal attention is sufficiently low to allow for late entry.

#### 4.2.4. INTER-MODAL ATTENTION ANALYSIS

To determine layers viable for skipping, we analyze inter-modal attention in addition to redundancy. If inter-modality interaction is not minimal, then even if one modality has significant redundancy, its output is still necessary for processing in the other modality, so skipping is not viable. In Figures 4–6, we use the VAR from Jiang et al. (2025) to visualize the self-attention between vision and text. Specifically, VAR for each head $h$ at layer $\ell$ for the $k$th text token $\mathbf{y}_k$ is defined as

$$\mathrm{VAR}^{(\ell)}(\mathbf{y}_k) \triangleq \sum_{j=0}^{h} \sum_{i=1}^{n} \mathbf{A}_k^{(\ell,j)}(a_k, i) \qquad (3)$$

where $\mathbf{A}_k^{(\ell,j)}(a_k, i)$ is the head-wise sum of the attention weights of the newly generated token $\mathbf{y}_k$ assigned to the image token $\mathbf{v}_i$. We visualize just the VAR with respect to the answer token. From the plots, in general across all datasets and models, the early and late layers have minimal cross attention according to this metric compared to the middle layers. Previous work has proposed this happens because the majority of visual information processing happens in these layers (Lin et al., 2025; Shang et al., 2025; Choraria et al., 2026; Jiang et al., 2025).

### 4.3. Connecting conditions to performance degradation

Based on our results in Section 4.2 and Theorems 1–4, we deduce that the early and late layers are often highly redundant with respect to visual information. Therefore, for the vision tokens, we skip the first $i$ layers (late entry) and the last $j$ layers (early stopping). We then compare the model performance against whether or not the redundancy conditions are being met, for each choice of skipping. The model forward pass is run while doing Late Entry and Early Exit at varying layers to showcase both positive and negative cases of conditions being met. This experiment aims to verify that our proposed theoretical conditions are good indicators for predicting model performance with skipping.

We begin with analyzing Late Entry layer skipping in Table 1, finding that while geometric and proximal redundancy are similar for the two LLaVA models for multimodal reasoning and VQA tasks, they noticeably differ from other models on the Captioning task. This implies that redundancy for Late Entry layer skipping is model and task dependent.

For the Early Exit layer skipping, in Table 2, the VAR across models differs, however minimally changes across tasks. Notably, however, VAR scores in the later layers on Captioning is much higher than the VQA/Mulimodal reasoning tasks. Notice that in both paradigms, performance decays when redundancy and attention conditions are not met.

---

**Key Insight #2**

Early layer textual redundancy is low in discriminative VQA tasks, but occurs after the Captioning tasks (See Figures 16 and 17 in the appendix). VAR scores in the later layers are much higher on Captioning tasks than discriminative VQA. This indicates that the task itself influences model redundancy.

---

### 4.4. Practical implications of this framework

This framework can be used to justify why and when certain layer skipping methods. However, using an entire dataset to find redundancy is a drawback. Therefore, we also pro-

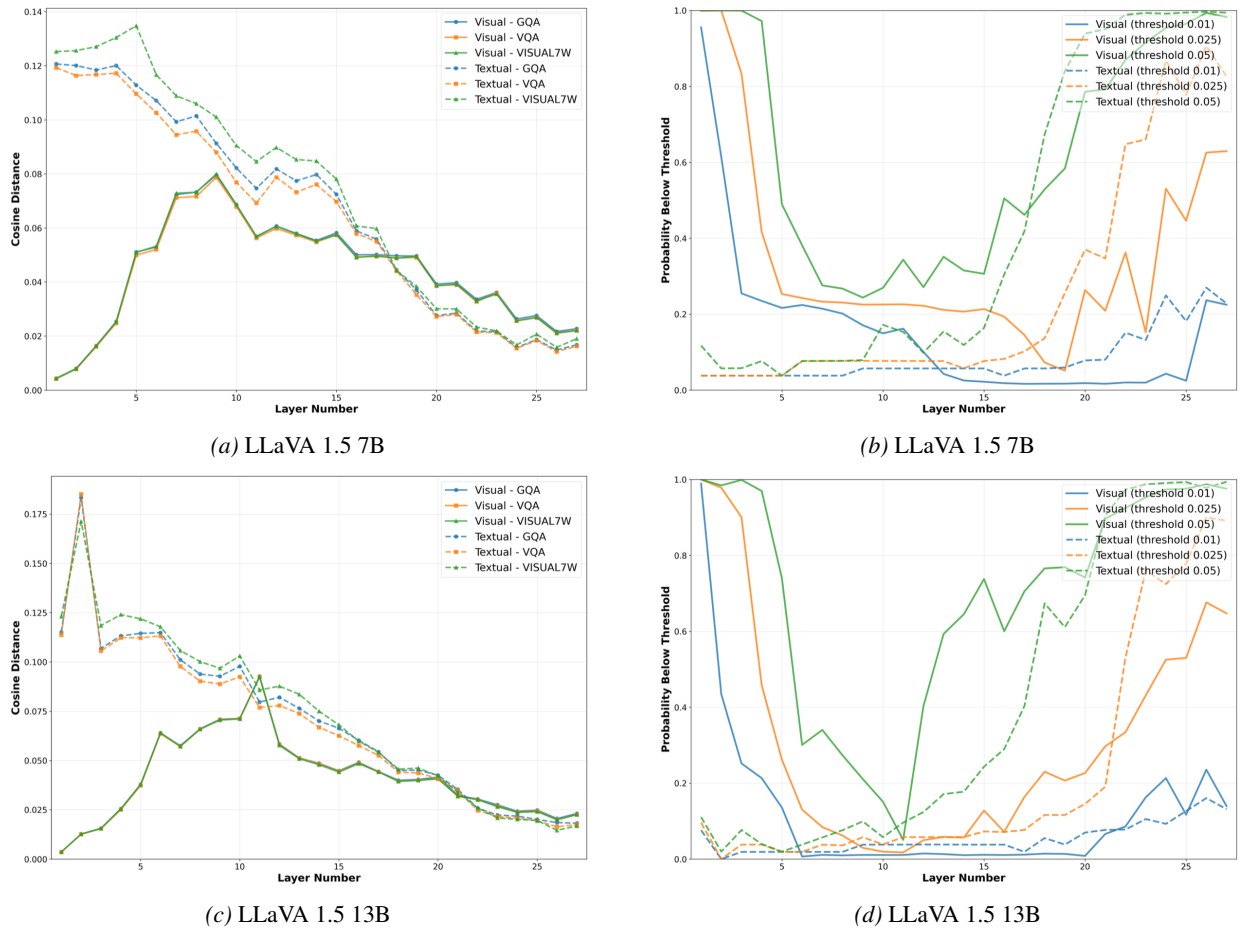

*Figure 3.* Empirical geometric and proximal redundancy experiments across layers for the LLaVA 1.5 7B/13B. Across all the general VQA tasks (see Table 3), models, and varying cosine distance threshold, the early layer vision tokens have low adjacent token cosine distances, and the textual and visual tokens have low adjacent token cosine distances in later layers. To keep the visualization clean, experiments on other tasks and models are moved to Appendix C.

| Task | Metric | LLaVA 1.5 7B | | | | LLaVA NeXT 7B Mistral | | | | LLaVA 1.5 13B | | | | Qwen 2.5 VL | | | |
|---|---|---|---|---|---|---|---|---|---|---|---|---|---|---|---|---|---|
| | | 0 | 4 | 8 | 12 | 0 | 4 | 8 | 12 | 0 | 4 | 8 | 12 | 0 | 4 | 8 | 12 |
| General VQA | $\rho(V_\ell, V_{\ell-1})$ | – | 0.025 | 0.073 | 0.060 | – | 0.006 | 0.037 | 0.070 | – | 0.025 | 0.066 | 0.058 | – | 0.0323 | 0.050 | 0.0353 |
| | $\mathbb{P}[D_V < 0.05]$ | – | 0.972 | 0.267 | 0.271 | – | 0.974 | 0.791 | 0.334 | – | 0.965 | 0.273 | 0.403 | – | 0.9246 | 0.625 | 0.922 |
| | Accuracy | **0.564** | **0.553** | 0.370 | 0.261 | **0.770** | **0.721** | 0.630 | 0.514 | **0.782** | **0.779** | 0.747 | 0.5010 | **0.823** | 0.587 | 0.579 | 0.526 |
| Text/Doc VQA | $\rho(V_\ell, V_{\ell-1})$ | – | 0.020 | 0.061 | 0.058 | – | 0.007 | 0.039 | 0.037 | – | 0.019 | 0.059 | 0.059 | – | 0.506 | 0.0692 | 0.0444 |
| | $\mathbb{P}[D_V < 0.05]$ | – | 0.992 | 0.378 | 0.337 | – | 0.967 | 0.891 | 0.410 | – | 0.995 | 0.362 | 0.411 | – | 0.6515 | 0.337 | 0.70 |
| | Accuracy | **0.5790** | **0.5640** | 0.5130 | 0.4920 | **0.703** | 0.5400 | 0.4420 | 0.3320 | **0.6920** | **0.6780** | 0.6260 | 0.6000 | **0.804** | 0.710 | 0.549 | 0.513 |
| Multimodal Reasoning | $\rho(V_\ell, V_{\ell-1})$ | – | 0.025 | 0.072 | 0.059 | – | 0.008 | 0.036 | 0.067 | – | 0.024 | 0.066 | 0.058 | – | 0.044 | 0.071 | 0.045 |
| | $\mathbb{P}[D_V < 0.05]$ | – | 0.965 | 0.333 | 0.322 | – | 0.961 | 0.821 | 0.346 | – | 0.978 | 0.314 | 0.441 | – | 0.731 | 0.335 | 0.692 |
| | Accuracy | **0.325** | **0.321** | 0.302 | 0.233 | **0.384** | 0.248 | 0.258 | 0.155 | **0.341** | **0.343** | 0.316 | 0.245 | **0.637** | 0.412 | 0.401 | 0.325 |
| | | 0 | 8 | 12 | 16 | 0 | 8 | 12 | 16 | 0 | 8 | 12 | 16 | 0 | 8 | 12 | 16 |
| Captioning | $\rho(V_\ell, V_{\ell-1})$ | – | 0.027 | 0.0729 | 0.0603 | – | 0.006 | 0.0372 | 0.0710 | – | 0.027 | 0.0669 | 0.0589 | – | 0.035 | 0.0519 | 0.0369 |
| | $\mathbb{P}[D_V < 0.05]$ | – | 0.951 | 0.267 | 0.270 | – | 0.976 | 0.782 | 0.314 | – | 0.948 | 0.265 | 0.386 | – | 0.906 | 0.587 | 0.897 |
| | BLEU | **0.217** | **0.270** | **0.261** | **0.221** | **0.168** | **0.242** | **0.158** | **0.181** | **0.199** | 0.182 | **0.222** | **0.214** | **0.141** | 0.112 | 0.121 | 0.067 |
| | CIDEr | **0.819** | **0.902** | **0.872** | 0.720 | **0.583** | **0.766** | 0.521 | **0.596** | **0.526** | 0.464 | **0.651** | **0.678** | **0.395** | **0.374** | 0.303 | 0.155 |
| | SPICE | **0.213** | **0.215** | **0.197** | 0.163 | **0.171** | **0.179** | **0.161** | 0.147 | **0.200** | 0.186 | **0.207** | 0.189 | **0.166** | **0.1402** | **0.146** | 0.092 |

*Table 1.* Summary of results for *Late Entry* on vision tokens. In this table, we only focus on the first 16 layers of each model. After this point, skipping causes each model to degrade even further. We split up the Multimodal Reasoning Task into datasets in which the models have accuracy close to guessing in the dataset. This was done to show how redundancy is less useful when a model has very poor performance, such that there is no model degradation. In this table, accuracy in bold indicates closeness to the baseline performance.

vide empirical insights into the generalizability of adjacent layer cosine distance in Appendix D and a theoretical functional redundancy generalization bound in Theorem 7 in

Appendix A. We show that while only considering a subset of examples, our framework still accurately measures model redundancy. Leveraging this theorem, we provide pseudo

| Task | Metric | LLaVA 1.5 7B | | | LLaVA NeXT 7B Mistral | | | LLaVA 1.5 13B | | | | | Qwen 2.5 VL 7B | | |
|---|---|---|---|---|---|---|---|---|---|---|---|---|---|---|---|
| | | 20 | 24 | 28 | 20 | 24 | 28 | 20 | 24 | 28 | 32 | 36 | 20 | 24 | 28 |
| General VQA | VAR | 4.33 | 4.11 | 2.44 | 2.53 | 1.98 | 2.19 | 4.26 | 3.27 | 2.64 | 2.58 | 4.01 | 8.43 | 6.65 | 3.45 |
| | Accuracy | **0.582** | **0.584** | **0.581** | **0.647** | **0.647** | **0.647** | 0.739 | **0.787** | **0.787** | **0.780** | **0.787** | **0.853** | **0.853** | **0.874** |
| Text/Doc VQA | VAR | 5.95 | 5.63 | 2.93 | 2.73 | 2.44 | 2.39 | 5.38 | 4.11 | 3.40 | 3.42 | 5.05 | 6.44 | 5.81 | 3.42 |
| | Accuracy | **0.592** | **0.594** | **0.595** | **0.684** | **0.701** | **0.701** | **0.691** | **0.710** | **0.708** | **0.708** | **0.708** | 0.883 | 0.882 | **0.912** |
| Multimodal Reasoning | VAR | 3.00 | 2.60 | 1.97 | 2.33 | 1.93 | 1.92 | 2.75 | 2.13 | 1.67 | 2.01 | 3.14 | 1.49 | 1.05 | 1.01 |
| | Accuracy | **0.322** | **0.325** | **0.326** | 0.343 | 0.341 | 0.342 | 0.313 | **0.335** | **0.335** | **0.335** | **0.333** | 0.533 | 0.530 | **0.593** |
| | | 20 | 24 | 28 | 20 | 24 | 28 | 16 | 20 | 24 | 28 | 32 | 20 | 24 | 28 |
| Captioning | VAR | 7.35 | 6.24 | 3.21 | 3.01 | 2.27 | 2.62 | 5.87 | 4.29 | 2.72 | 4.00 | 5.74 | 8.49 | 3.56 | 1.32 |
| | BLEU | **0.203** | **0.219** | **0.216** | **0.165** | **0.168** | **0.168** | **0.143** | **0.161** | **0.184** | **0.194** | **0.143** | 0.069 | **0.107** | **0.121** |
| | CIDEr | 0.573 | **0.620** | **0.625** | **0.561** | **0.560** | **0.566** | 0.377 | 0.433 | **0.499** | **0.509** | **0.511** | 0.185 | **0.320** | **0.358** |
| | SPICE | **0.206** | **0.213** | **0.213** | **0.163** | **0.164** | **0.165** | 0.160 | 0.176 | **0.193** | **0.195** | **0.197** | 0.094 | **0.138** | **0.152** |

*Table 2.* Summary of results for *Early Exit* on vision tokens. In this table, we only focus on the layers starting on the 20th layer of each model. Before this point, early skipping causes significant performance degradation across models due to being to visual information still being processed, which is supported in Figures 4–6. We include layer 20 to see how are framework is applicable when VAR is moderate). We split up the Multimodal Reasoning Task into datasets in which the models have accuracy close to guessing in the dataset. This was done to show how redundancy is less useful when a model has very poor performance, such that there is no model degradation. In this table, accuracy in bold indicates closeness to the baseline performance.

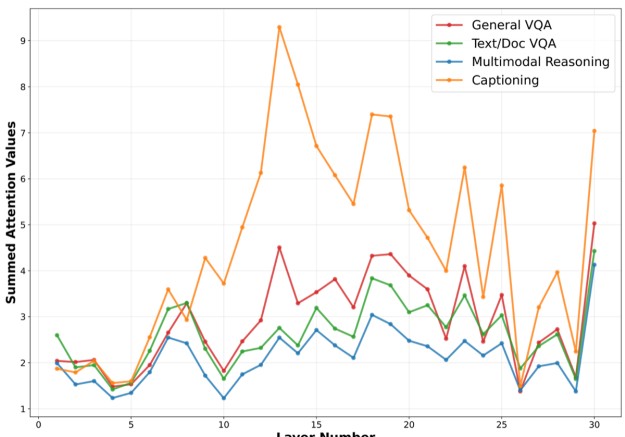
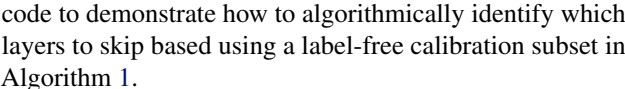

*Figure 4.* Visual Attention Ratio with respect to the answer text token for LLaVA 1.5 7B. In general, the vision-to-text attention is mainly in the middle layers, with the early and late layers having minimal vision-to-text attention.

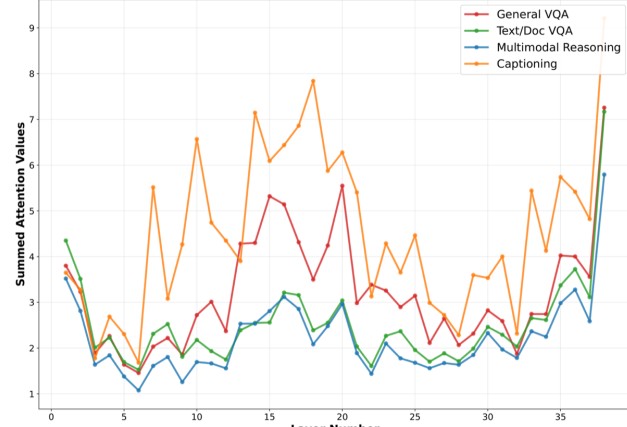

*Figure 5.* Visual Attention Ratio with respect to the answer text token for LLaVA 1.5 13B. In general, the vision-to-text attention is mainly in the middle layers, with the early and late layers having minimal vision-to-text attention.

code to demonstrate how to algorithmically identify which layers to skip based using a label-free calibration subset in Algorithm 1.

## 5. Discussion

Our redundancy framework is grounded in probability and information theory, making it general enough to provide a unified foundation for analyzing a wide range of pruning techniques. To illustrate its flexibility, we show how it connects to several existing methods. The layer-level redundancy claims of Shukor & Cord (2024) find theoretical justification through our framework, as does Skip-Vision (Zeng et al., 2025), which merges tokens based on cosine similarity rankings—a strategy whose justification aligns closely with our functional redundancy theorems. Similarly, FlexiDepth (Luo et al., 2025a) can be interpreted as learning

$\mathbb{E}[Z \mid X_\ell]$ at each layer $\ell$, while ST$^3$ (Zhuang et al., 2025) identifies "lazy" layers that contribute minimally beyond their predecessors using cosine similarity. Our framework applies directly to all of these approaches, offering a common theoretical lens for understanding why they work.

The task type itself influences when VLMs have redundancy. Following the results of (Jiang et al., 2025; Kaduri et al., 2025), visual information processing is done predominately in the middle layers of the model. Additionally, (Hartman et al., 2025) showed that vision tokens are likely copied in the early layers. Therefore, we hypothesize that textual tokens are initially processed in the early model layers to understand what information is needed from the visual tokens in the middle layers. In our results, we find that captioning behaves differently from VQA. Specifically, captioning tasks are less amenable to early exit skipping. This per-

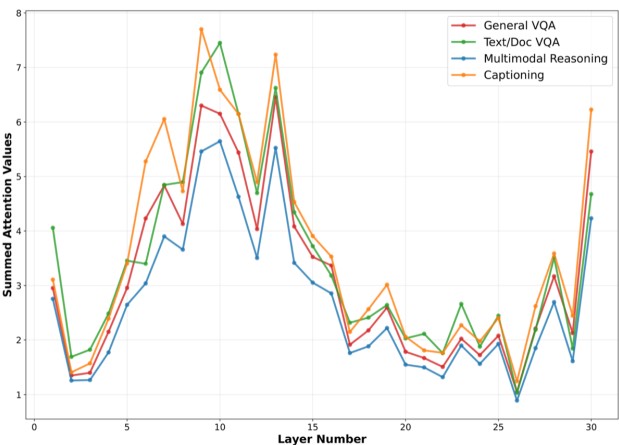

*Figure 6.* Visual Attention Ratio with respect to the answer text token for LLaVA NeXT 7B Mistral. In general, the vision-to-text attention is mainly in the middle layers, with the early and late layers having minimal vision-to-text attention.

---

**Algorithm 1** Finding Skippable Layers

---

**Require:** VLM $\Phi$ with $N$ layers, calibration subset $D'$, and geometric redundancy threshold $\epsilon_g$, proximal redundancy distance threshold $t$, proximal redundancy probability threshold $\alpha$, and visual attention ratio threshold $\tau$.

**Ensure:** Late Entry index $L$, Early Exit index $j$.

1: *Compute task-specific metrics for all layers* $l \in \{1, \dots, N\}$
2: **for** each layer $l = 1 \to N$ **do**
3:    $\mathcal{D}_l \leftarrow$ Expected adjacent layer cosine distance over $D'$.
4:    $\mathbb{P}_l \leftarrow$ Expected probability of $t$-proximal states over $D'$.
5:    $V_l \leftarrow$ Expected Visual Attention Ratio (VAR) over $D'$.
6: **end for**
7: *Determine Late Entry point $L$*
8: Search for the largest layer index $l \in \{1, \dots, N\}$ such that for every preceding layer $k \leq l$:
9:    $\mathcal{D}_k < \epsilon_g$ AND $\mathbb{P}_k \geq \alpha$.
10: $L \leftarrow \max\{l \mid \text{Condition 2 is satisfied}\}$.
11: *Determine Early Exit point $j$*
12: Search for the smallest layer index $l \in \{1, \dots, N\}$ such that for every subsequent layer $k \geq l$:
13:    $\mathcal{D}_k < \epsilon_g$ AND $\mathbb{P}_k \geq \alpha$ AND $V_k < \tau$.
14: $j \leftarrow \min\{l \mid \text{Condition 3 is satisfied}\}$.
15: **Return** Late Entry layer $L$, Early Exit layer $j$.

---

formance difference stems from functional requirements: captioning tasks are more demanding than VQA. Drawing from results in Assran et al. (2023), captioning requires high-fidelity representations to ensure grounded, complex descriptions. Conversely, VQA allows early "collapse" of high-dimensional input into narrow, task-specific features. For example, answering a specific question about an object's color requires only a narrow scope of information, making the subsequent layers functionally redundant and thus more viable to an early exit. In our framework, the "task variable" for captioning represents a comprehensive scene synthesis, while for VQA, it is a targeted semantic feature extracted much earlier in the forward pass.

## 6. Conclusion & Future Work

In this work, we propose a theoretical framework for studying redundancy in VLMs. We define a general notion of layer skipping that encompasses many existing techniques, providing a unified lens for understanding when and why they succeed. Our framework yields easy-to-compute conditions that can identify redundant layers using only a small subset of unlabeled data, without requiring downstream task performance. We validate this empirically by showing that our conditions predict when layer skipping is viable across different models and datasets, and that skipping non-redundant layers leads to significant degradation.

A natural direction for future work is understanding why these redundancies arise, whether as a byproduct of vision-language pretraining strategies or as an intentional mechanism for multimodal processing. Beyond this, the general nature of our framework suggests it may extend to other multimodal settings such as audio-language or video-language models. Investigating whether redundancy patterns change predictably with model scale could also inform the design of more efficient architectures.

## Acknowledgments

This research used the DeltaAI advanced computing and data resource, which is supported by the National Science Foundation (award OAC 2320345) and the State of Illinois. DeltaAI is a joint effort of the University of Illinois Urbana-Champaign and its National Center for Supercomputing Applications.

## Impact Statement

By understanding when existing layer skipping techniques can be applied successfully without model degradation, our paper motivates the development of improved techniques. Layer skipping techniques reduce the overhead of compute needed for inference on VLMs. Our paper introduces theory which better motives why these techniques work in practice. Which in turn motivates work which reduces the number of tokens per layer, thereby decreasing the inference time of VLMs. This has very important applications to edge computing, which is becoming more and more prevalent as

privacy of foundation models becomes a growing concern. Our work directly contributes to a theoretical understanding of redundancy in multimodal models.

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

# Appendix

## A. Additional Framework Details

### A.1. Proofs of Lemmas

*Lemma* 1. Let $(X, d)$ be a metric space and $x, y, z \in X$. Then $d(x,y)^2 \leq 2d(x,z)^2 + 2d(z,y)^2$.

*Proof.* Follows from the triangle inequality and the AM-GM inequality. $\square$

*Lemma* 2. Suppose $X, Y$ are random vectors with unit norm (with probability 1). Let $\rho(x,y) = 1 - \frac{\langle x,y \rangle}{\|x\|\|y\|}$ be the cosine distance between $x$ and $y$. Then $\mathbb{E}[\rho(X,Y)] < \frac{\epsilon}{2}$ implies $\mathbb{E}[\|X - Y\|_2^2] < \epsilon$.

*Proof.* Observe that

$$\mathbb{E}[\|X - Y\|_2^2] = \mathbb{E}[\|X\|^2 + \|Y\|^2 - 2\langle X, Y \rangle] = 2\mathbb{E}[1 - \langle X, Y \rangle] \tag{4}$$
$$= 2\mathbb{E}[\rho(X, Y)] \tag{5}$$
$$< \epsilon. \tag{6}$$

$\square$

*Lemma* 3. Let $(X, \langle \cdot, \cdot \rangle)$ be a real inner product space and $a, b, c \in X$. Then $\|a + b + c\|^2 \leq 3(\|a\|^2 + \|b\|^2 + \|c\|^2)$.

*Proof.* Expanding $\|a + b + c\|^2$ we get $\|a\|^2 + \|b\|^2 + \|c\|^2 + 2\langle a, b \rangle + 2\langle b, c \rangle + 2\langle a, c \rangle$. Thus we want to show that $0 \leq 2\|a\|^2 + 2\|b\|^2 + 2\|c\|^2 - 2\langle a, b \rangle - 2\langle b, c \rangle - 2\langle a, c \rangle$. This is equivalent to showing $0 \leq \|a - b\|^2 + \|b - c\|^2 + \|a - c\|^2$ and since $\| \cdot \|^2$ is non-negative, we are done. $\square$

*Lemma* 4. Suppose $Y$—$X$—$Z$ is a Markov chain. Then $\mathbb{E}_{(X,Y)}[D(p_{Z|X} \| p_{Z|Y})] = I(Z; X|Y)$.

*Proof.* Observe that

$$\mathbb{E}[D(p_{Z|X} \| p_{Z|Y})] = \mathbb{E}\left[\int p_{Z|X}(z|X) \log \frac{p_{Z|X}(z|X)}{p_{Z|Y}(z|Y)} dz\right] \tag{7}$$
$$= \mathbb{E}_{(X,Y),Z \sim p(\cdot|X)}\left[\log \frac{p(Z|X)}{p(Z|Y)}\right] \tag{8}$$
$$= \mathbb{E}_{(X,Y),Z \sim p(\cdot|X,Y)}\left[\log \frac{p(Z|X,Y)}{p(Z|Y)}\right] \tag{9}$$
$$= \mathbb{E}_{X,Y,Z}\left[\log \frac{p(Z,X|Y)}{p(X|Y)p(Z|Y)}\right] \tag{10}$$
$$= I(Z; X|Y) \tag{11}$$

where (9) follows from the Markov property and (11) is the definition of conditional mutual information. $\square$

### A.2. Proofs of Propositions and Theorems

*Proposition* 1. Let $\rho : \mathcal{X} \times \mathcal{X} \to \mathbb{R}$ be a symmetric function with $0 \leq \rho \leq 1$. Then

$$\mathbb{P}[\rho(X, Y) > t] < \frac{\epsilon - t}{1 - t} \quad \text{implies} \quad \mathbb{E}[\rho(X, Y)] < \epsilon$$

*Proof.* Define the random variable $D := \rho(X, Y)$. By the tail integration formula,

$$\mathbb{E}[D] = \int_0^1 \mathbb{P}[D > s]ds \tag{12}$$

$$= \int_0^t \mathbb{P}[D > s]ds + \int_t^1 \mathbb{P}[D > s]ds \tag{13}$$

$$\leq t \cdot 1 + (1-t)\mathbb{P}[D > t] \tag{14}$$

$$< t + (1-t)(\frac{\epsilon - t}{1 - t}) \tag{15}$$

$$= \epsilon. \tag{16}$$

$\square$

*Theorem* 1. Let $X_\ell, X_{\ell-1}$ be unit-norm random variables and $Z$ be another random variable (e.g. hidden representations of layers $\ell, \ell - 1$ and the task ground truth respectively). Let $\rho(x, y) = 1 - \frac{\langle x, y \rangle}{\|x\|\|y\|}$. Assume $\mathbb{E}[\rho(X, Y)] < \frac{\epsilon}{2}$ and that

$$h(x, y) = E[Z|X_\ell = x, X_{\ell-1} = y]$$

is $\alpha$-Lipschitz in the first argument and $\beta$-Lipschitz in the second. Then $E[\|E[Z|X_\ell] - E[Z|X_{\ell-1}]\|_2^2] < 2(\alpha^2 + \beta^2)\epsilon$.

*Proof.* Observe that

$$\mathbb{E}[Z|X_\ell] - E[Z|X_{\ell-1}] = \mathbb{E}[\mathbb{E}[Z|X_\ell, X_{\ell-1}]|X_\ell] - \mathbb{E}[\mathbb{E}[Z|X_\ell, X_{\ell-1}]|X_{\ell-1}] \tag{17}$$

$$= \mathbb{E}[h|X_\ell] - \mathbb{E}[h|X_{\ell-1}] \tag{18}$$

By Lemma 2 we have that $\mathbb{E}[\|X_\ell - X_{\ell-1}\|_2^2] < \epsilon$ since $\|X_\ell\|$ and $\|X_{\ell-1}\|$ are unit-norm.
By Lemma 1 we have

$$(\mathbb{E}[h|X_\ell] - \mathbb{E}[h|X_{\ell-1}])^2 \leq 2(\mathbb{E}[h|X_\ell] - h)^2 + 2(\mathbb{E}[h|X_{\ell-1}] - h)^2.$$

Furthermore, since expectation is order-preserving, we have

$$\mathbb{E}[(\mathbb{E}[h|X_\ell] - \mathbb{E}[h|X_{\ell-1}])^2] \leq 2\mathbb{E}[(\mathbb{E}[h|X_\ell] - h)^2] + 2\mathbb{E}[(\mathbb{E}[h|X_{\ell-1}] - h)^2] \tag{19}$$

$$= 2\mathbb{E}[\text{Var}(h|X_\ell)] + 2\mathbb{E}[\text{Var}(h|X_{\ell-1})]. \tag{20}$$

By definition,

$$\mathbb{E}[\text{Var}(h(X_\ell, X_{\ell-1})|X_\ell = x)] = \mathbb{E}[\mathbb{E}[(h(X_\ell, X_{\ell-1}) - \mathbb{E}[h(X_\ell, X_{\ell-1})|X_\ell = x])^2|X_\ell = x]] \tag{21}$$

$$\leq \mathbb{E}[\mathbb{E}[(h(X_\ell, X_{\ell-1}) - h(X_\ell, X_{\ell-1} = \mathbb{E}[X_{\ell-1}|X_\ell = x]))^2|X_\ell = x]] \tag{22}$$

$$\leq \beta^2\mathbb{E}[\mathbb{E}[\|X_{\ell-1} - \mathbb{E}[X_{\ell-1}|X_\ell = x]\|_2^2|X_\ell = x]] \tag{23}$$

$$= \beta^2\mathbb{E}[\|X_{\ell-1} - \mathbb{E}[X_{\ell-1}|X_\ell = x]\|_2^2] \tag{24}$$

$$\leq \beta^2\mathbb{E}[\|X_{\ell-1} - X_\ell\|_2^2] \tag{25}$$

$$< \beta^2\epsilon \tag{26}$$

where (22) holds by the optimality of the minimum mean squared error (MMSE) estimator, (23) holds by the Lipschitz assumption, (24) holds by the tower property (Law of Total Expectation), and (25) holds by the optimality of the MMSE estimator once again. By symmetry, the same holds $X_{\ell-1}$ (i.e. $\mathbb{E}[\text{Var}(h|X_{\ell-1} = y)] < \alpha^2\epsilon$) so $\mathbb{E}[(\mathbb{E}[Z|X_\ell] - \mathbb{E}[Z|X_{\ell-1}])^2] \leq 2(\alpha^2 + \beta^2)\epsilon$ $\square$

*Theorem* 2. Let $X_\ell, X_{\ell-1}$ be unit-norm random variables and $Z$ be another random variable (e.g. layer activations of layers $\ell, \ell - 1$ and a task variable respectively). Let $\rho(x, y) = 1 - \frac{\langle x, y \rangle}{\|x\|\|y\|}$. Assume $\mathbb{E}[\rho(X, Y)] < \frac{\epsilon}{2}$ and that

$$h(x, y) = E[Z|X_\ell = x, X_{\ell-1} = y]$$

is $\alpha$-Lipschitz in the first argument and $\beta$-Lipschitz in the second. Let $\hat{f}_\ell$ be a finite-sample estimate of $f_\ell^*(x) = \mathbb{E}[Z|X_\ell = x]$ and $\hat{f}_{\ell-1}(x)$ be a finite-sample estimate of $f_{\ell-1}^*(x) = \mathbb{E}[Z|X_{\ell-1} = x]$. Further let $\eta_\ell = \mathbb{E}[\|\hat{f}_\ell(X_\ell) - f_\ell^*(X_\ell)\|^2]$ and $\eta_{\ell-1} = \mathbb{E}[\|\hat{f}_{\ell-1}(X_{\ell-1}) - f_{\ell-1}^*(X_{\ell-1})\|^2]$. We then have

$$\mathbb{E}[\|\hat{f}_\ell(X_\ell) - \hat{f}_{\ell-1}(X_{\ell-1})\|^2] < 3\eta_\ell + 3\eta_{\ell-1} + 6(\alpha^2 + \beta^2)\epsilon.$$

*Proof.* By Lemma 2 we have that $\mathbb{E}[\|X_\ell - X_{\ell-1}\|_2^2] < \epsilon$.

Observe that

$$\hat{f}_\ell(X_\ell) - \hat{f}_{\ell-1}(X_{\ell-1}) = (-f_\ell^*(X_\ell) + \hat{f}_\ell(X_\ell)) + (f_{\ell-1}^*(X_{\ell-1}) - \hat{f}_{\ell-1}(X_{\ell-1})) + (f_\ell^*(X_\ell) - f_{\ell-1}^*(X_{\ell-1})).$$

Thus by Lemma 3 we have:

$$\|\hat{f}_\ell(X_\ell) - \hat{f}_{\ell-1}(X_{\ell-1})\|^2 \leq 3(\|f_\ell^*(X_\ell) - \hat{f}_\ell(X_\ell)\|^2 + \|f_{\ell-1}^*(X_{\ell-1}) - \hat{f}_{\ell-1}(X_{\ell-1})\|^2 + \|f_{\ell-1}^*(X_{\ell-1}) - f_\ell^*(X_\ell)\|^2).$$

Finally, by taking expectations, we get

$$\mathbb{E}[\|\hat{f}(X_\ell) - \hat{f}(X_{\ell-1})\|^2] \leq 3(\eta_\ell + \eta_{\ell-1} + 2(L_1^2 + L_2^2)\epsilon) = 3\eta_\ell + 3\eta_{\ell-1} + 6(\alpha^2 + \beta^2)\epsilon.$$

$\square$

*Theorem* 3 (Continuous Fano's Inequality; (Duchi & Wainwright, 2013)). Let $X_\ell, X_{\ell-1}$ be unit-norm vectors over the support $\mathcal{X}$. Define

$$\overline{\mathbb{B}}_\rho(t) = \{x' \in \mathbb{R}^d | \rho(x, x') \leq t\}.$$

Let $\mu$ be the Lebesgue measure. Assume $\mu(\partial\mathcal{X})$ and $\sup_{x \in \mathcal{X}} \mu(\partial(\mathbb{B}_\rho(t) \cap \mathcal{X}))$ are finite where the Lebesgue measure is taken over their respective dimensions. Let $P_t = \mathbb{P}[\rho(X_\ell, X_{\ell-1}) \geq t]$. Then if $X_\ell$ is uniform over $\mathcal{X}$,

$$I(X_\ell, X_{\ell-1}) \geq (1 - P_t)\log\left(\frac{\mu(\mathcal{X})}{\sup_{x \in \mathcal{X}} \mu(\mathbb{B}_\rho(t) \cap \mathcal{X})}\right) - \log 2.$$

*Proof.* Observe that $X_\ell$—$X_{\ell-1}$—$X_{\ell-1}$ is trivially a Markov chain. The result follows from applying results from Duchi & Wainwright (2013). $\square$

*Theorem* 4 ((Braun & Pokutta, 2015)). Let $X_\ell, X_{\ell-1}$ be unit-norm random variables with shared support $\mathcal{X}$. Let $\rho : \mathcal{X} \times \mathcal{X} \to \mathbb{R}$ be a symmetric function (e.g. a metric). Let $\overline{B}(t, x) := \{x' \in \mathcal{X} | \rho(x, x') \leq t\}$, $P_t = \mathbb{P}[\rho(X_{\ell-1}, X_\ell) > t]$. Let

$$p_{min} := \inf_{x \in \mathcal{X}} \mathbb{P}[(X_{\ell-1}, x) \in \overline{B}(t, x)] \quad \text{and} \quad p_{max} := \sup_{x \in \mathcal{X}} \mathbb{P}[(X_{\ell-1}, x) \in \overline{B}(t, x)]$$

with $0 \leq p_{min} < 1$ and $0 < p_{max} \leq 1$ and $p_{min} + p_{max} < 1$. Then,

$$I(X_{\ell-1}; X_\ell) \geq (1 - P_t)\log\frac{1}{p_{max}} - P_t\log(1 - p_{min}) - H_2(P_t).$$

*Proof.* Follows directly from Proposition 2.2 in (Braun & Pokutta, 2015) with $R = \{(x, x') \in \mathcal{X} \times \mathcal{X} : \rho(x, x') \leq t\}$. $\square$

*Corollary* 4 (Conditional Entropy Fano's Inequality; (Braun & Pokutta, 2015)). With the same conditions and notations as Theorem 4 we have

$$H(X_\ell|X_{\ell-1}) \leq H(X_\ell) + \log p_{max} + H(P_t) + P_t\log\frac{1 - p_{min}}{p_{max}}.$$

*Proof.* Follows from Corollary 2.3 in (Braun & Pokutta, 2015) with $R = \{(x, x') \in \mathcal{X} \times \mathcal{X} : \rho(x, x') \leq t\}$. $\square$

*Theorem* 5. Suppose there are random variables $Z, X_\ell, X_{\ell-1}$ with $Z \in \mathbb{R}^d$ and $X_\ell, X_{\ell-1}$ continuous unit-norm random variables. Further suppose $\|Z\|_2 \leq B$ almost surely and that $X_{\ell-1}$— $X_\ell$— $Z$ is a Markov chain. Then $\mathbb{E}[\|\mathbb{E}[Z|X_\ell] - \mathbb{E}[Z|X_{\ell-1}]\|_2^2] \leq 2B^2 I(Z; X_\ell|X_{\ell-1})$.
If, in addition, there exists finite $C$ such that $H(X_\ell|Z, X_{\ell-1}) \geq -C$ then $\mathbb{E}[\|\mathbb{E}[Z|X_\ell] - \mathbb{E}[Z|X_{\ell-1}]\|_2^2] \leq 2B^2(H(X_\ell|X_{\ell-1}) + C)$. In particulary, if $X_\ell$ is discrete then $C = 0$ and if $p_{X_\ell|Z, X_{\ell-1}}(x) \leq M \, \forall x$ then $C = \log M$.

*Proof.* Fix an $a, b$ and consider the conditional probability distributions $p_{Z|X_\ell=a}$ and $p_{Z|X_{\ell-1}=b}$. We then have that

$$\mathbb{E}[Z|X_\ell = a] - \mathbb{E}[Z|X_{\ell-1} = b] = \int_{\mathbb{R}^d} z(p_{Z|X_\ell=a}(z) - p_{Z|X_{\ell-1}=b}(z))dz.$$

Thus,

$$\|\mathbb{E}[Z|X_\ell = a] - \mathbb{E}[Z|X_{\ell-1} = b]\|_2 = \left\|\int_{\mathbb{R}^d} z(p_{Z|X_\ell=a}(z) - p_{Z|X_{\ell-1}=b}(z))dz\right\|_2 \tag{27}$$

$$\leq \int_{\mathbb{R}^d} \|z\|_2 |p_{Z|X_\ell=a}(z) - p_{Z|X_{\ell-1}=b}(z)|dz \tag{28}$$

$$\leq B \int_{\mathbb{R}^d} |p_{Z|X_\ell=a}(z) - p_{Z|X_{\ell-1}=b}(z)|dz \tag{29}$$

$$= 2B\delta_{TV}(p_{Z|X_\ell=a}, p_{Z|X_{\ell-1}=b}) \tag{30}$$

where (28) holds by the triangle inequality. Thus we have,

$$\mathbb{E}[\|\mathbb{E}[Z|X_\ell] - \mathbb{E}[Z|X_{\ell-1}]\|_2^2] \leq 4B^2\mathbb{E}[\delta_{TV}(p_{Z|X_\ell=a}, p_{Z|X_{\ell-1}=b})^2] \tag{31}$$

$$\leq 2B^2\mathbb{E}[D(p_{Z|X_\ell}||p_{Z|X_{\ell-1}})]. \tag{32}$$

where (32) holds by Pinsker's inequality.

Now, by Lemma 4 we have $2B^2\mathbb{E}[D(p_{Z|X_\ell}||p_{Z|X_{\ell-1}})] = 2B^2 I(Z; X_\ell|X_{\ell-1})$. Finally, we know that $I(Z; X_\ell|X_{\ell-1}) = H(X_\ell|X_{\ell-1}) - H(X_\ell|Z, X_{\ell-1}) \leq H(X_\ell|X_{\ell-1}) + C$. Thus,

$$\mathbb{E}[\|\mathbb{E}[Z|X_\ell] - \mathbb{E}[Z|X_{\ell-1}]\|_2^2] \leq 2B^2 H(X_\ell|X_{\ell-1}) + C.$$

$\square$

*Theorem* 6 (Late Entry). Let $f = f^n \circ f^{n-1} \circ \cdots \circ f^1$ be the $n$ layers in a VLM. Fix a layer $L \in \{1, 2, \ldots, n\}$. Let

$$X_l = \begin{pmatrix} V_l \\ T_l \end{pmatrix} = \begin{pmatrix} f^l_{vis}(V_{l-1}, T_{l-1}) \\ f^l_{text}(V_{l-1}, T_{l-1}) \end{pmatrix}$$

be the true states with $X_0 = (V, T)^T$. Let $\phi = f^n \circ \cdots \circ f^{L+1}$ be the "tail" of the VLM.

Let $Y_{true} = f(X)$ and $Y_{skip} = \phi\left(\begin{pmatrix} V_1 \\ \tilde{T}_L \end{pmatrix}\right)$, where the approximated text sequence is defined recursively as $\tilde{T}_0 = T_0$ and $\tilde{T}_l = f^l_{text}(0, \tilde{T}_{l-1})$ for $l > 0$.

Assume $\phi$ is $\mu$-Lipschitz with respect to the $\ell_2$-norm, and $f^l_{text}$ is $\lambda$-Lipschitz in the second argument. Further assume that $\|V_i - V_{i+1}\| \leq \epsilon$ for all $i$, and the visual dependency bound is $\|f^l_{text}(0, T) - f^l_{text}(V_{l-1}, T)\| \leq \delta$. Then:

$$\|Y_{true} - Y_{skip}\| \leq \mu\left((L-1)\epsilon + \delta\left(\frac{\lambda^L - 1}{\lambda - 1}\right)\right).$$

*Proof.* By repeated application of the triangle inequality, we have $\|V_1 - V_L\| \leq \sum_{i=1}^{L-1} \|V_i - V_{i+1}\| \leq (L-1)\epsilon$.

Next, we bound the text error. Define $E_l := \|T_l - \tilde{T}_l\|$. For $l \geq 1$:

$$E_l = \|f^l_{text}(V_{l-1}, T_{l-1}) - f^l_{text}(0, \tilde{T}_{l-1})\| \tag{33}$$

$$\leq \|f^l_{text}(V_{l-1}, T_{l-1}) - f^l_{text}(0, T_{l-1})\| + \|f^l_{text}(0, T_{l-1}) - f^l_{text}(0, \tilde{T}_{l-1})\| \tag{34}$$

$$\leq \delta + \lambda\|T_{l-1} - \tilde{T}_{l-1}\| \tag{35}$$

$$= \delta + \lambda E_{l-1}. \tag{36}$$

With the base case $E_0 = 0$, this linear recurrence has the closed form solution:

$$E_L \leq \delta \sum_{k=0}^{L-1} \lambda^k = \delta\left(\frac{\lambda^L - 1}{\lambda - 1}\right).$$

Finally, bounding the total error:

$$\|Y_{true} - Y_{skip}\| = \left\| \phi \begin{pmatrix} V_L \\ T_L \end{pmatrix} - \phi \begin{pmatrix} V_1 \\ \tilde{T}_L \end{pmatrix} \right\| \tag{37}$$

$$\leq \mu \left\| \begin{pmatrix} V_L - V_1 \\ T_L - \tilde{T}_L \end{pmatrix} \right\|_2 \tag{38}$$

$$= \mu \sqrt{\|V_L - V_1\|^2 + \|T_L - \tilde{T}_L\|^2}. \tag{39}$$

Using the inequality $\sqrt{a^2 + b^2} \leq a + b$ for non-negative $a, b$, we obtain:

$$\|Y_{true} - Y_{skip}\| \leq \mu \left( \|V_L - V_1\| + \|T_L - \tilde{T}_L\| \right) \tag{40}$$

$$\leq \mu \left( (L-1)\epsilon + \delta \left( \frac{\lambda^L - 1}{\lambda - 1} \right) \right). \tag{41}$$

$\square$

*Lemma* 5 (Early Exit). Let $f$ and $X_l$ be defined as in Theorem 6, with $n$ total layers. Fix a vision-exit layer $L \in \{1, 2, \ldots, n-1\}$. Let $Y_{true} = X_n = \begin{pmatrix} V_n \\ T_n \end{pmatrix}$ be the true final state. Let $Y_{skip} = \begin{pmatrix} V_L \\ \tilde{T}_n \end{pmatrix}$ be the approximated final state where visual computation halts at layer $L$. The approximated text sequence continues through layer $n$, defined recursively for $l > L$ as $\tilde{T}_l = f_{text}^l(V_L, \tilde{T}_{l-1})$, with the base case $\tilde{T}_L = T_L$.

Assume that visual features minimally update after layer $L$ such that $\|V_i - V_{i-1}\| \leq \epsilon_v$ for all $i > L$. Assume $f_{text}^l$ is $\lambda$-Lipschitz in the second argument. Further, assume the late cross-modal dependency is bounded by $\|f_{text}^l(V_{l-1}, T) - f_{text}^l(V_L, T)\| \leq \delta_v$ for all $l > L$. Then:

$$\|Y_{true} - Y_{skip}\| \leq (n-L)\epsilon_v + \delta_v \left( \frac{\lambda^{n-L-1} - 1}{\lambda - 1} \right).$$

*Proof.* Similar to Theorem 6, we bound the distance between the true final state and the vision-exited state by independently bounding the vision and text errors.

First, by repeated application of the triangle inequality on the residual vision updates, we bound the vision error:

$$\|V_n - V_L\| \leq \sum_{i=L+1}^{n} \|V_i - V_{i-1}\| \tag{42}$$

$$\leq (n-L)\epsilon_v. \tag{43}$$

Next, we bound the compounding text error. Define $E_l := \|T_l - \tilde{T}_l\|$. For the first skipped layer ($l = L + 1$), since both the true and approximated states utilize $V_L$ and $T_L$, the error is exactly zero:

$$E_{L+1} = \|f_{text}^{L+1}(V_L, T_L) - f_{text}^{L+1}(V_L, \tilde{T}_L)\| \tag{44}$$

$$= 0. \tag{45}$$

For subsequent layers $l \geq L + 2$, the vision inputs diverge. We bound the text error using the triangle inequality:

$$E_l = \|f_{text}^l(V_{l-1}, T_{l-1}) - f_{text}^l(V_L, \tilde{T}_{l-1})\| \tag{46}$$

$$\leq \|f_{text}^l(V_{l-1}, T_{l-1}) - f_{text}^l(V_L, T_{l-1})\| + \|f_{text}^l(V_L, T_{l-1}) - f_{text}^l(V_L, \tilde{T}_{l-1})\| \tag{47}$$

$$\leq \delta_v + \lambda \|T_{l-1} - \tilde{T}_{l-1}\| \tag{48}$$

$$= \delta_v + \lambda E_{l-1}. \tag{49}$$

Since $E_{L+1} = 0$, we are accumulating the $\delta_v$ error over $(n - L - 1)$ steps. This linear recurrence yields the closed-form sum:

$$E_n \leq \delta_v \sum_{k=0}^{n-L-2} \lambda^k = \delta_v \left( \frac{\lambda^{n-L-1} - 1}{\lambda - 1} \right).$$

Finally, bounding the total state error using the previously established inequality $\sqrt{a^2 + b^2} \leq a + b$:

$$\|Y_{true} - Y_{skip}\| = \left\| \begin{pmatrix} V_n - V_L \\ T_n - \tilde{T}_n \end{pmatrix} \right\|_2 \tag{50}$$

$$\leq \|V_n - V_L\| + \|T_n - \tilde{T}_n\| \tag{51}$$

$$\leq (n - L)\epsilon_v + \delta_v \left( \frac{\lambda^{n-L-1} - 1}{\lambda - 1} \right). \tag{52}$$

$\square$

*Theorem* 7. Let $X_\ell, X_{\ell-1}, Z$ be continuous random variables and $R(\omega) = \|\mathbb{E}[Z|X_\ell(\omega)] - \mathbb{E}[Z|X_{\ell-1}(\omega)]\|_2^2$ with $0 \leq R \leq B$ almost surely. Let $D' = \{\omega_1, \omega_2, \ldots, \omega_n\}$ be $n$ independent and identically distributed samples. Let $\mu = \mathbb{E}[R(\omega)]$ and $\hat{\mu} = \frac{1}{n} \sum_{i=1}^n R(\omega_i)$. Then

$$\mathbb{P}[|\hat{\mu} - \mu| \geq t] \leq 2 \exp(-\frac{2nt^2}{B^2}).$$

*Proof.* Observe that the samples in $D'$ are i.i.d and that $R$ is bounded almost surely. Thus, one can apply Hoeffding's inequality to $R$ and establish the final bound. $\square$

Refer to Appendix D for an empirical study of the generalizability of our method.

### A.3. Connection to PID

For readers familiar with information theory, much of the vocabulary and intuition discussed regarding informational redundancy may sound very similar to notions of redundant and unique information in partial information decomposition (PID). PID (of 3 finite-support random variables) proposes that the mutual information $I(X; Y, Z)$ can be decomposed into the following terms.

1. Unique Information: *Uni*$(X : Y \backslash Z)$ and *Uni*$(X : Z \backslash Y)$ for the unique information that $Y$ contains about $X$ and $Z$ contains about $X$ respectively.

2. Redundant Information: *Red*$(X : Y, Z)$, which is the information about $X$ that both $Y$ and $Z$ share.

3. Synergistic Information (sometimes called Shared Information): *Syn*$(X : Y, Z)$, which is the information about $X$ that can only be derived from the combination of both $Y$ and $Z$.

The proposed decomposition of the mutual information is given by the following definition.

*Definition* 5 (Partial Information Decomposition).

$$I(X; Y, Z) \triangleq Uni(X : Y \backslash Z) + Uni(X : Z \backslash Y) + Red(X : Y, Z) + Syn(X : Y, Z)$$

$$I(X; Y) \triangleq Uni(X : Y \backslash Z) + Red(X : Y, Z)$$

$$I(X; Z) \triangleq Uni(X : Z \backslash Y) + Red(X : Y, Z).$$

In fact, the connection between PID and informational redundancy can be somewhat formalized through the following observation.

*Lemma* 6. Let $X, Y$ be discrete random variables. Then *Uni*$(X : X \backslash Y) = H(X|Y)$.

*Proof.* From the chain rule for mutual information we know that $I(X;Y|Z) = I(X;Y,Z) - I(X;Z)$. Using Definition 5 we see that $I(X;Y|Z) = Uni(X:Y\backslash Z) + Syn(X:Y,Z)$ and similarly, $I(X;Z|Y) = Uni(X:Z\backslash Y) + Syn(X:Y,Z)$. Thus $I(X;Y|X) = 0 = Uni(X:Y\backslash X) + Syn(X:X,Y)$. By the non-negativity of PID we get that $Uni(X:Y\backslash X) = Syn(X:X,Y) = 0$. Thus, $I(X;X|Y) = H(X|Y) = Uni(X:X\backslash Y)$. $\square$

Thus, if one considers $X = X_\ell, Y = X_{\ell-1}$, we recover our definition of informational redundancy using PID. This also offers a PID interpretation of redundancy: the unique information about the current layer, which only the current layer has, should be low. Further, since PID considers three random variables, this also allows us to consider a combination of our functional and informational redundancy by considering the quantity $Uni(Z:X_\ell\backslash X_{\ell-1})$. This would be the unique information that $X_\ell$ has about a target random variable $Z$ that $X_{\ell-1}$ does not have.

The unique information quantity $Uni(X:Y\backslash Z)$ also has a widely accepted definition given by Bertschinger et al. (2014), which is the solution to following convex optimzation problem.

*Definition* 6 (BROJA definition; (Bertschinger et al., 2014)).

$$Uni(X:Y\backslash Z) \triangleq \min_{Q\in\Delta_P} I_Q(X:Y|Z) \tag{53}$$

where $\Delta$ is the set of all joint distributions on $X, Y, Z$ and $\Delta_P = \{Q \in \Delta : Q(X = x, Y = y) = P(X = x, Y = y)$ and $Q(X = x, Z = z) = P(X = x, Z = z) \; \forall x \in \text{supp}(X), y \in \text{supp}(Y), z \in \text{supp}(Z)\}$. That is the set of distributions that agree on the marginals.

If one can bound this value, then one recovers a "functional information-theoretic redundancy".

### A.4. Estimation of Lipschitz Constant

In Theorems 1 and 2, we assume the Lipschitz continuity of the estimator. To see if the Lipschitz constants observed are small enough to make the conclusions from the theorems reasonable in practice, the Lipschitz constants of the layers in LLaVA 1.5 7B, LLaVA NeXT 7B Mistral, and Qwen 2.5 VL 7B were estimated via the spectral norm. A plot of the spectral norms can be seen in Figures 7, 8, and 9.

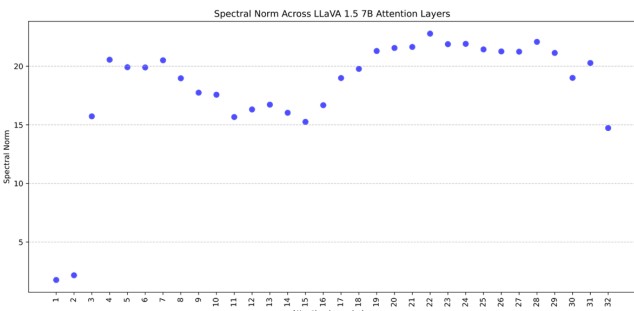

*Figure 7.* Spectral norm of all layers in LLaVA 1.5 7B. All spectral norms are relatively small (below 25).

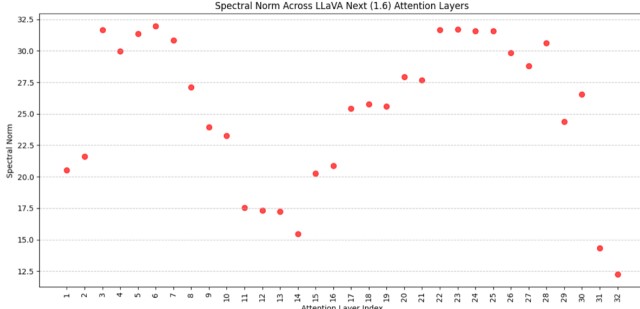

*Figure 8.* Spectral norm of all layers in LLaVA NeXT 7B Mistral. All spectral norms are relatively small.

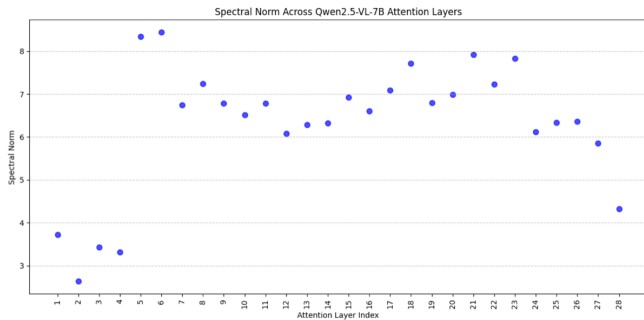

*Figure 9.* Spectral norm of all layers in Qwen 2.5 VL 7B. All spectral norms are relatively small.

| Task | Datasets |
|------|----------|
| General VQA | GQA, VQA, Visual7W |
| Text/Doc VQA | AI2D, OCRBench, TextVQA |
| Multimodal Reasoning | MMMU, RealWorldQA, MMStar, MathVision |
| Image Captioning | Coco, Flickr30k |

*Table 3.* Dataset Organization by task

## B. Datasets

In this work, we experiment on General Visual Question Answering (VQA), Text/Doc VQA, Multimodal Reasoning, and Math Reasoning. See the table below for a dataset breakdown.

### B.1. Evaluation method of each dataset

We split our datasets into two groups depending on whether they contain MCQ questions that can be answered in one token or not. The MCQ datasets include Visual7W, AI2D, MMMU, MMStar, and a subset of MathVision. We used an LLM-as-a-judge approach to evaluate the other datasets. These include: VQA, GQA, TextVQA, OCRBench, RealWorldQA, and a subset of MathVision.

For the MCQ datasets, we ran a forward pass to generate exactly the predicted letter (A, B, C, D). We then directly compared the predicted letter to the correct letter. Some of the datasets included Yes/No questions, and these were evaluated the same way.

For the LLM-as-a-judge datasets, we generated 256 tokens. We then used GPT-5 (OpenAI, 2025) to evaluate if the predicted answer was correct given the question and correct answer. This approach was beneficial to avoid association reasoning problems that smaller models (13B or less parameters) may have answering complex questions.

### B.2. Experimental Design Details

Hidden states are extracted **post-residual** addition. LLaVA 1.5 vision tokens are contained in a fixed contiguous block of 576 tokens, so the vision tokens are isolated by finding this block starting from the index of the image token. LLaVA NeXT, Qwen 2.5 VL, and Deepseek-VL use dynamic token counts, so we search for a contiguous block of image token ids.

## C. Further Results from Section 4.2

We include further experiments on General VQA, Text/Doc VQA, Multimodal Reasoning, and Captioning datasets on LLaVA 1.5 7B/13B, LLaVA NeXT 7B Mistral, DeepSeek VL 7B, and Qwen 2.5 VL 7B to validate our results are consistent across task types.

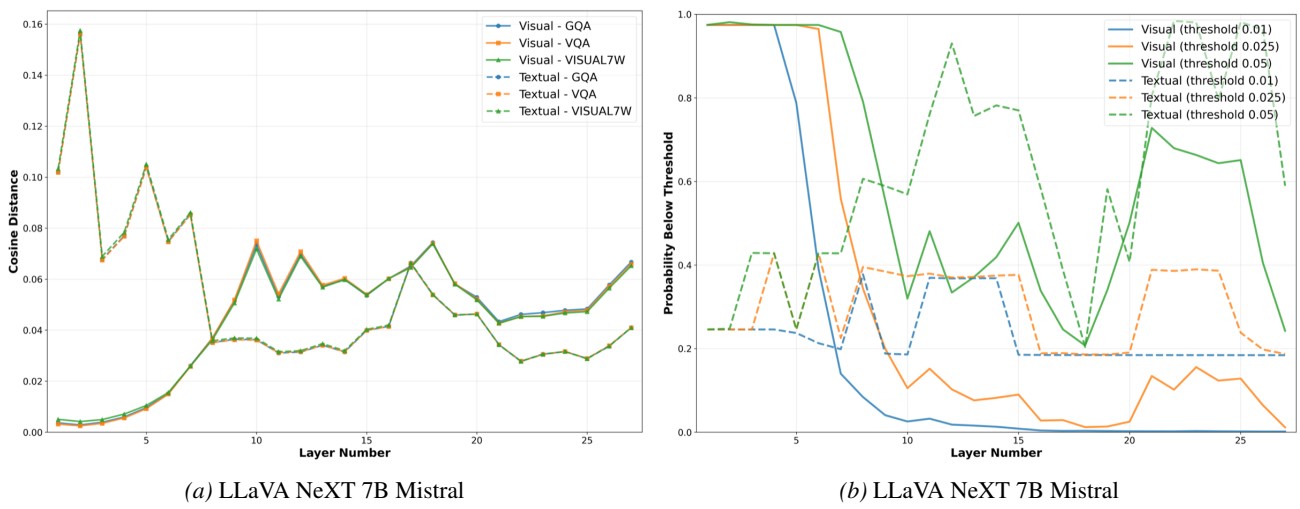

*(a)* LLaVA NeXT 7B Mistral

*(b)* LLaVA NeXT 7B Mistral

*Figure 10.* Empirical Geometric and Proximal Redundancy versus layer for LlaVA NeXT 7B Mistral. Across all datasets in the Text/Doc VQA task (see Table 3) and models, the early and late layer vision tokens have low adjacent token cosine distances.

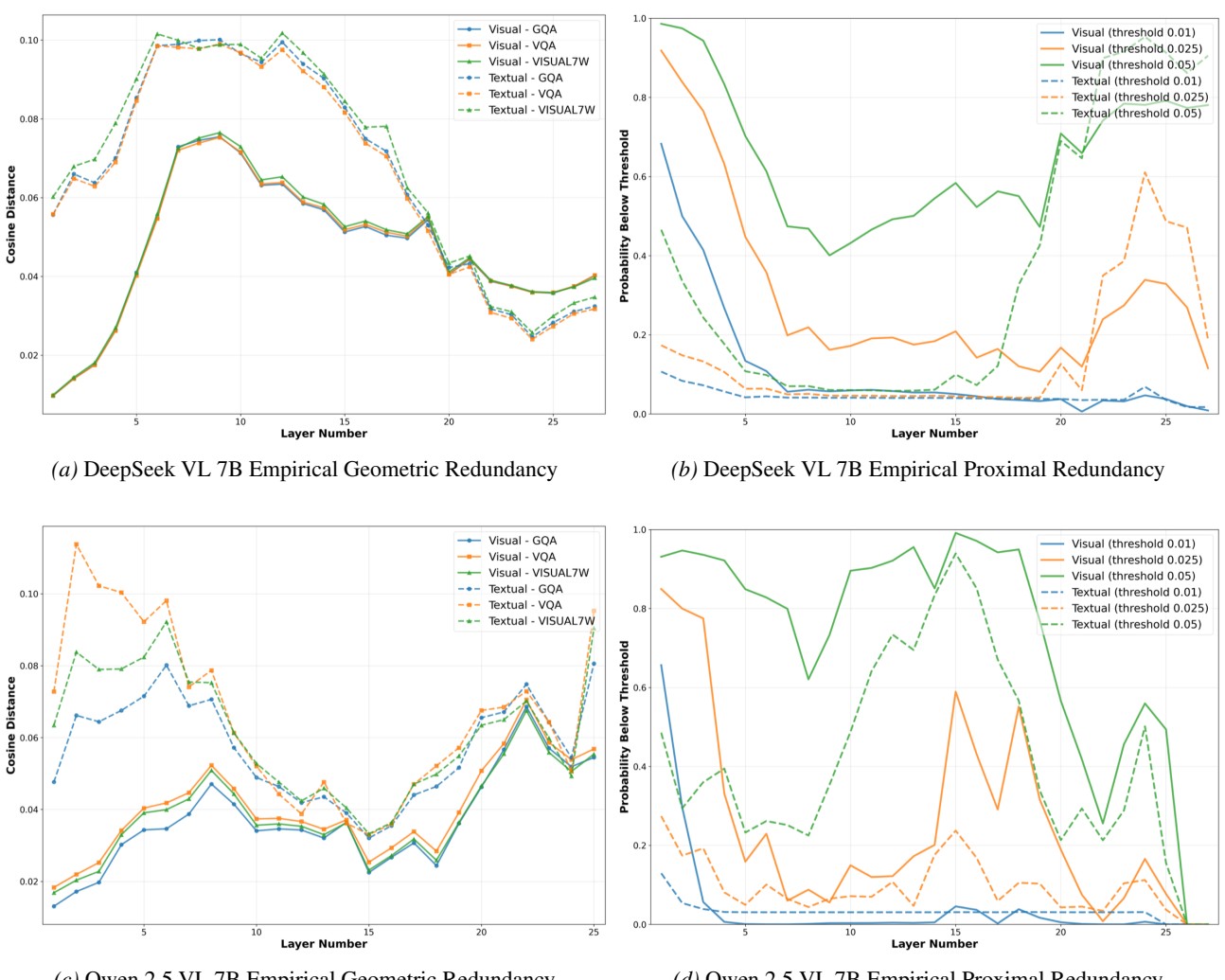

*(a)* DeepSeek VL 7B Empirical Geometric Redundancy

*(b)* DeepSeek VL 7B Empirical Proximal Redundancy

*(c)* Qwen 2.5 VL 7B Empirical Geometric Redundancy

*(d)* Qwen 2.5 VL 7B Empirical Proximal Redundancy

*Figure 11.* Empirical Geometric and Proximal Redundancy versus layer for the Qwen 2.5 VL and Deepseek VL 7B VLMs. Across all datasets in the General VQA task (see Table 3) and models, the early layer vision tokens have low adjacent token cosine distances, and the textual and visual tokens have low adjacent token cosine distances in later layers.

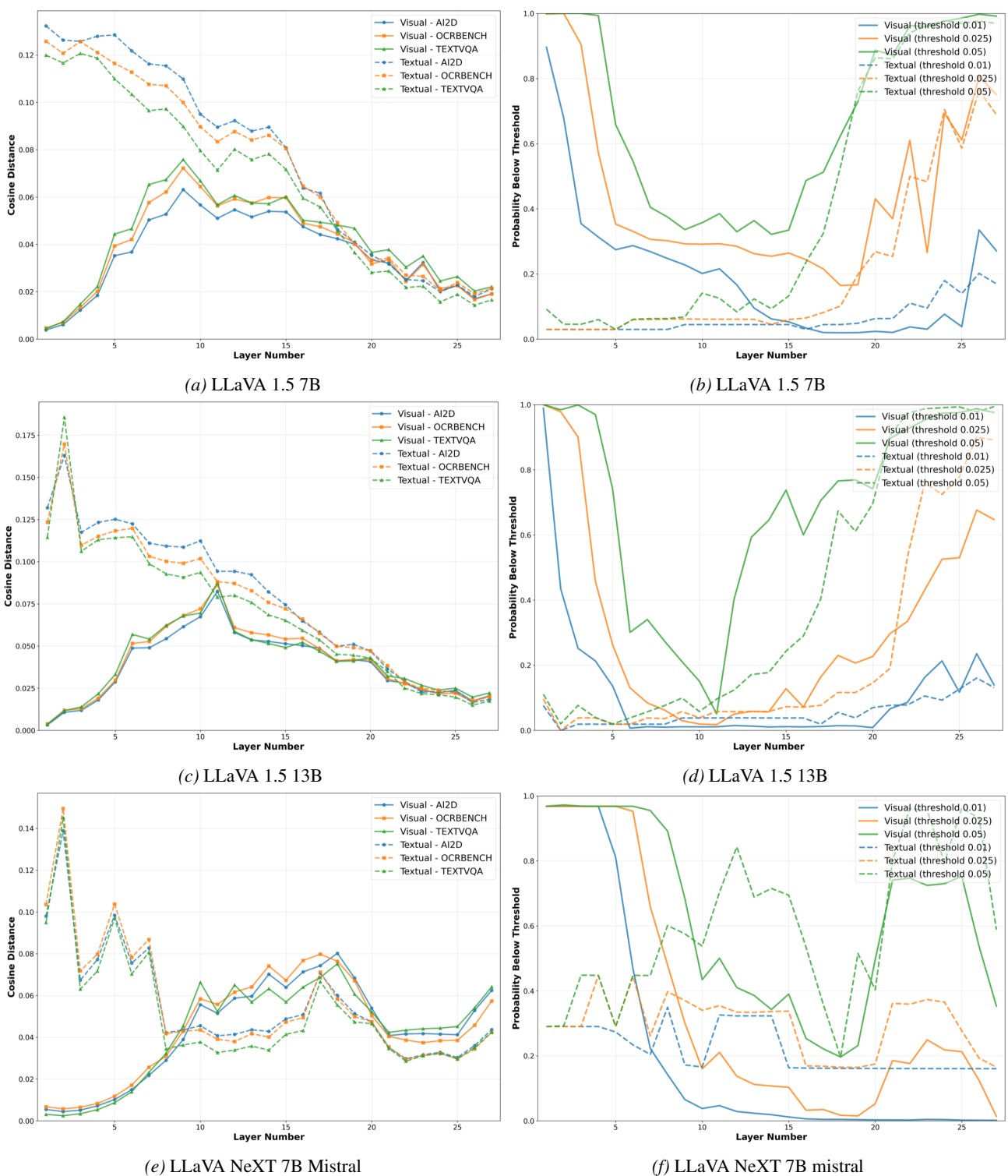

*(a)* LLaVA 1.5 7B

*(b)* LLaVA 1.5 7B

*(c)* LLaVA 1.5 13B

*(d)* LLaVA 1.5 13B

*(e)* LLaVA NeXT 7B Mistral

*(f)* LLaVA NeXT 7B mistral

*Figure 12.* Empirical Geometric and Proximal Redundancy versus layer for the LlaVA models. Across all datasets in the Text/Doc VQA task (see Table 3) and models, the early and late layer vision tokens have low adjacent token cosine distances.

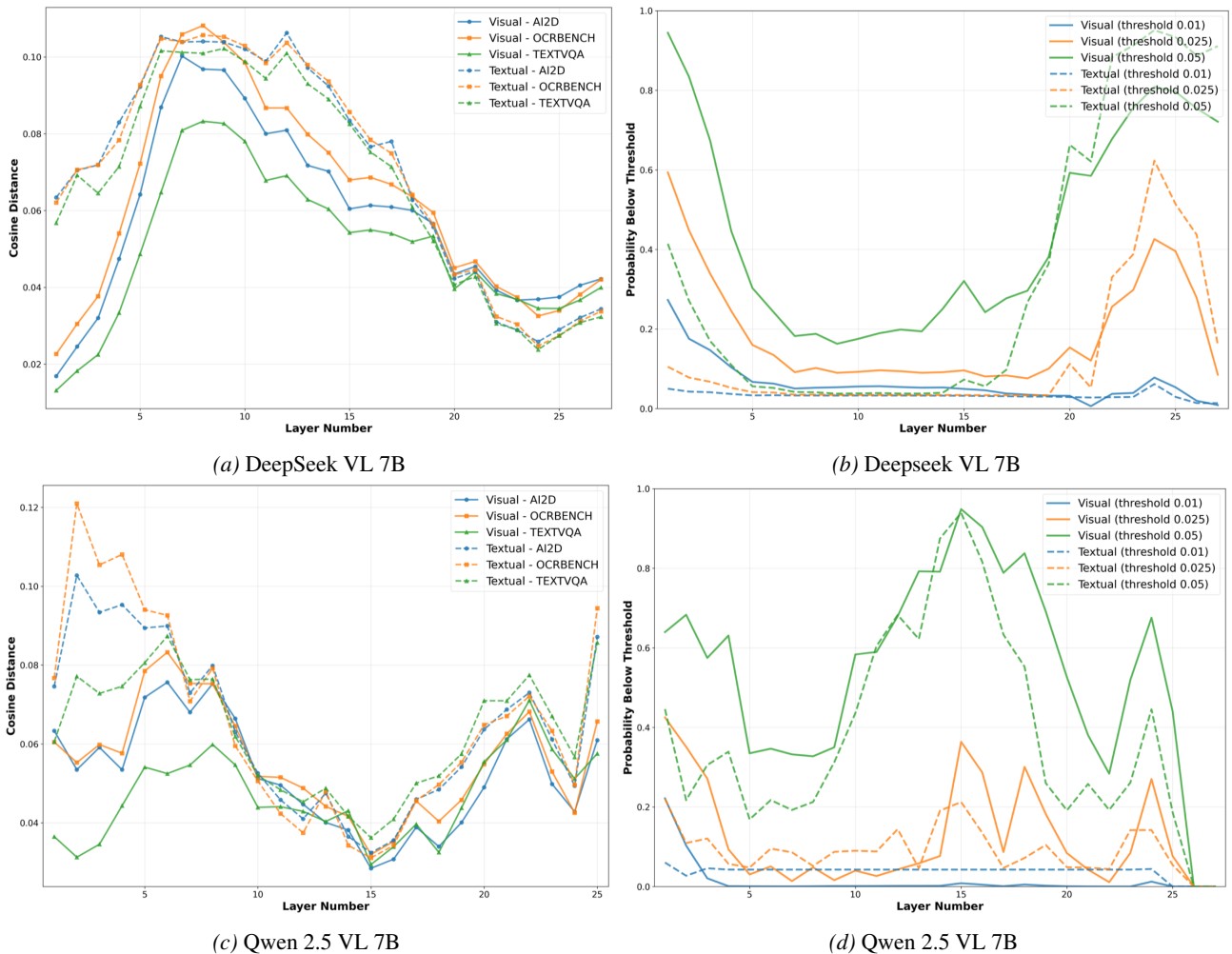

*(a)* DeepSeek VL 7B

*(b)* Deepseek VL 7B

*(c)* Qwen 2.5 VL 7B

*(d)* Qwen 2.5 VL 7B

*Figure 13.* Empirical Geometric and Proximal Redundancy versus layer for the Qwen 2.5 VL and Deepseek VL 7B VLMs. Across all datasets in the Text/Doc VQA task (see Table 3) and models, the early and late layer vision tokens have low adjacent token cosine distances.

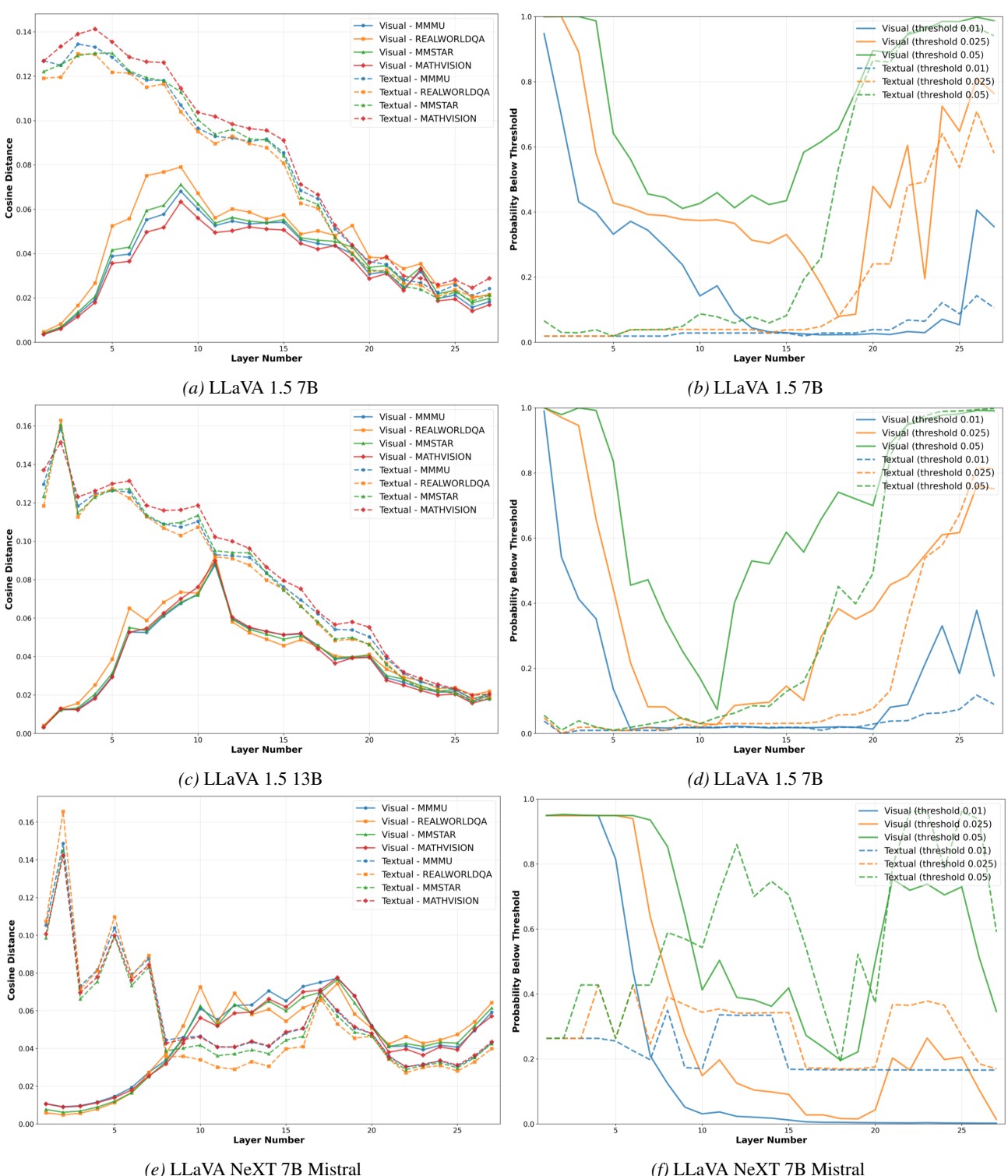

*(a)* LLaVA 1.5 7B

*(b)* LLaVA 1.5 7B

*(c)* LLaVA 1.5 13B

*(d)* LLaVA 1.5 7B

*(e)* LLaVA NeXT 7B Mistral

*(f)* LLaVA NeXT 7B Mistral

*Figure 14.* Empirical Geometric Redundancy and Proximal Redundancy between hidden states versus layer. Across all the Multimodal Reasoning task (see Table 3) on LLaVA 1.5/1.6, the early and late layer vision tokens have low adjacent token cosine distances.

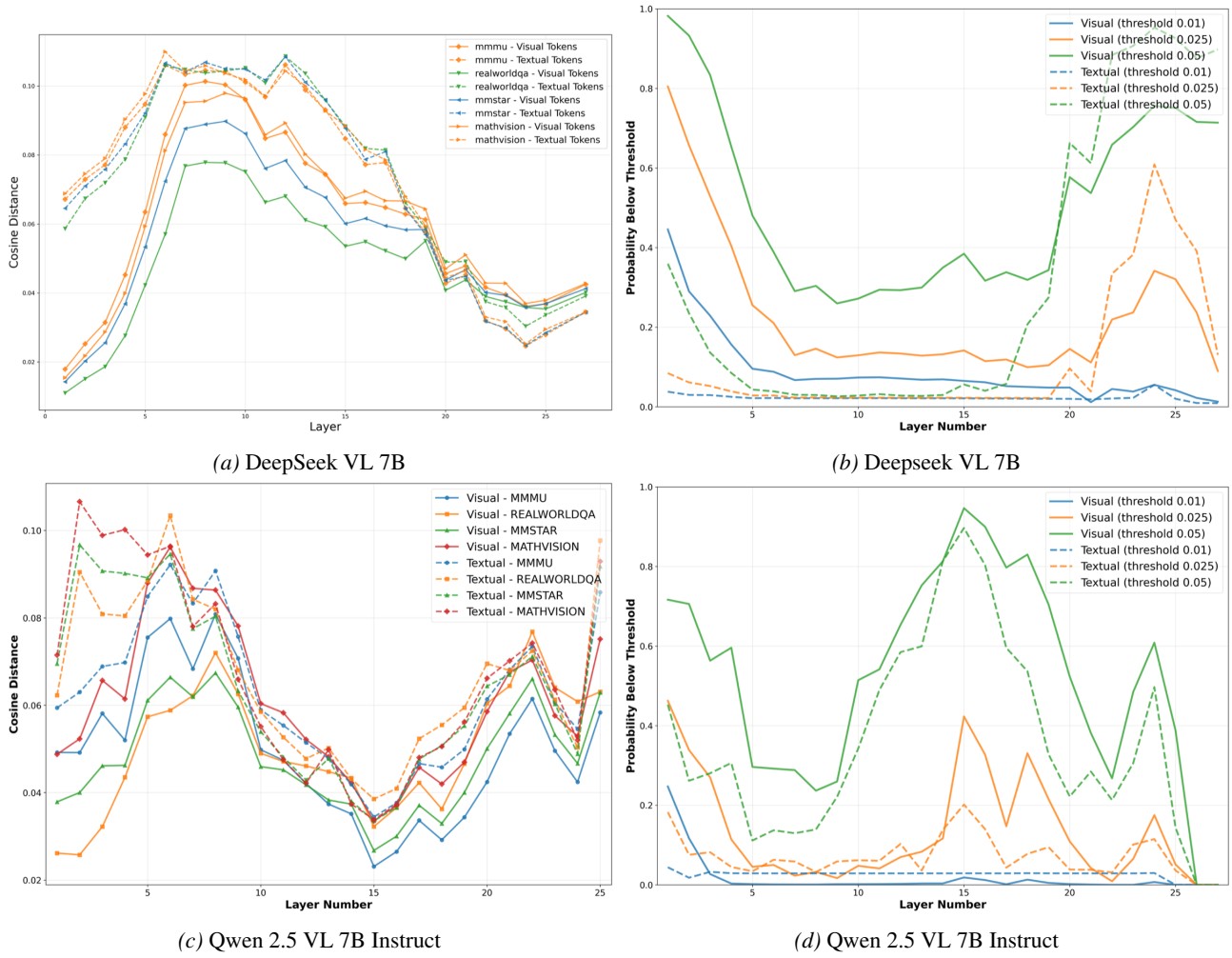

*(a)* DeepSeek VL 7B

*(b)* Deepseek VL 7B

*(c)* Qwen 2.5 VL 7B Instruct

*(d)* Qwen 2.5 VL 7B Instruct

*Figure 15.* Empirical Geometric and Proximal Redundancy versus layer. Across the Multimodal Reasoning task (see Table 3) on Qwen 2.5 VL Instruct and DeepSeek VL 7B, the early and late layer vision tokens have low adjacent token cosine distances.

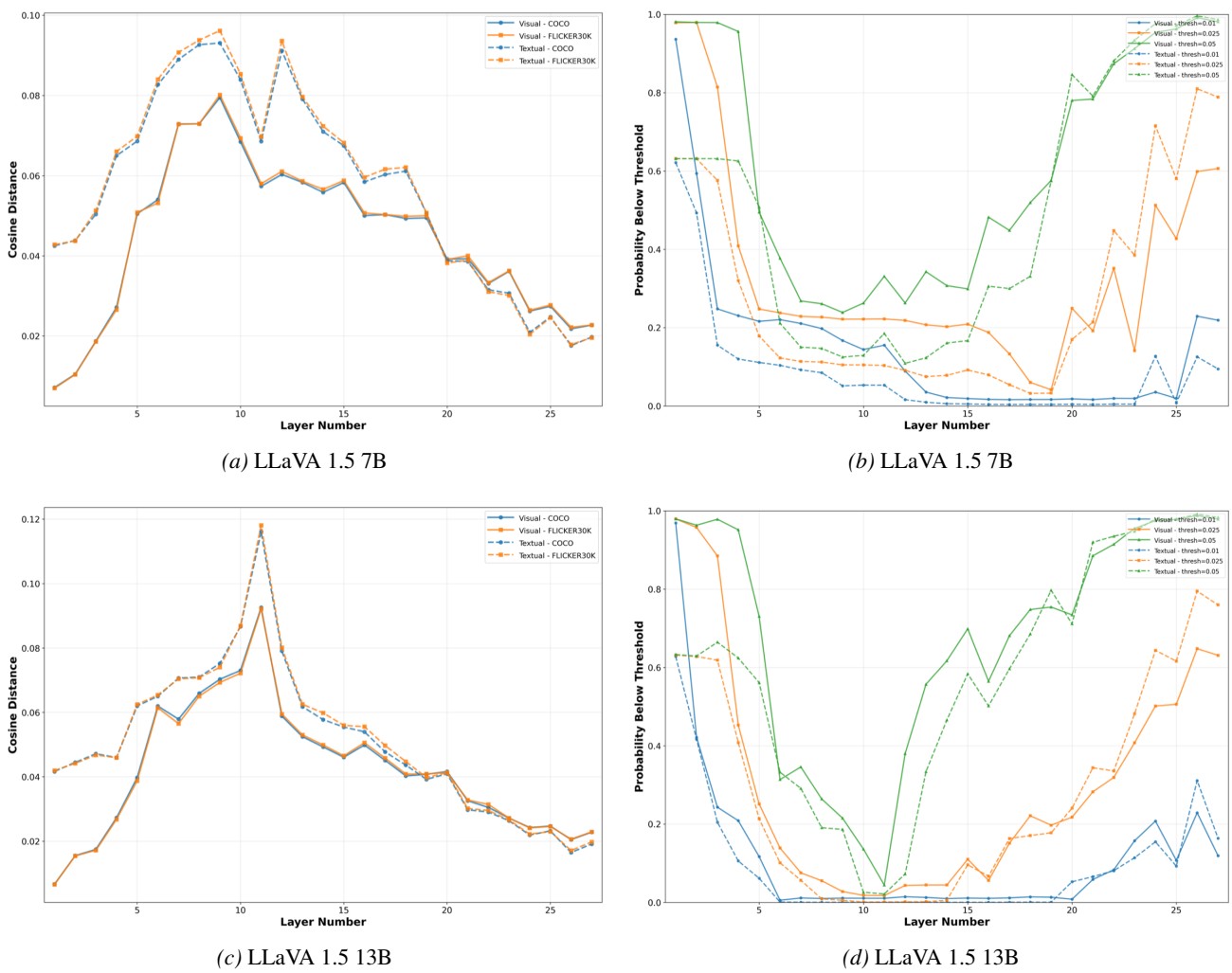

*(a)* LLaVA 1.5 7B

*(b)* LLaVA 1.5 7B

*(c)* LLaVA 1.5 13B

*(d)* LLaVA 1.5 13B

*Figure 16.* Empirical Geometric and Proximal Redundancy versus layer on LLaVA 1.5 architectures. Across the Captioning task (see Table 3) on LLaVA 1.5 7B and 13B, the early and late layer vision tokens have low adjacent token cosine distances.

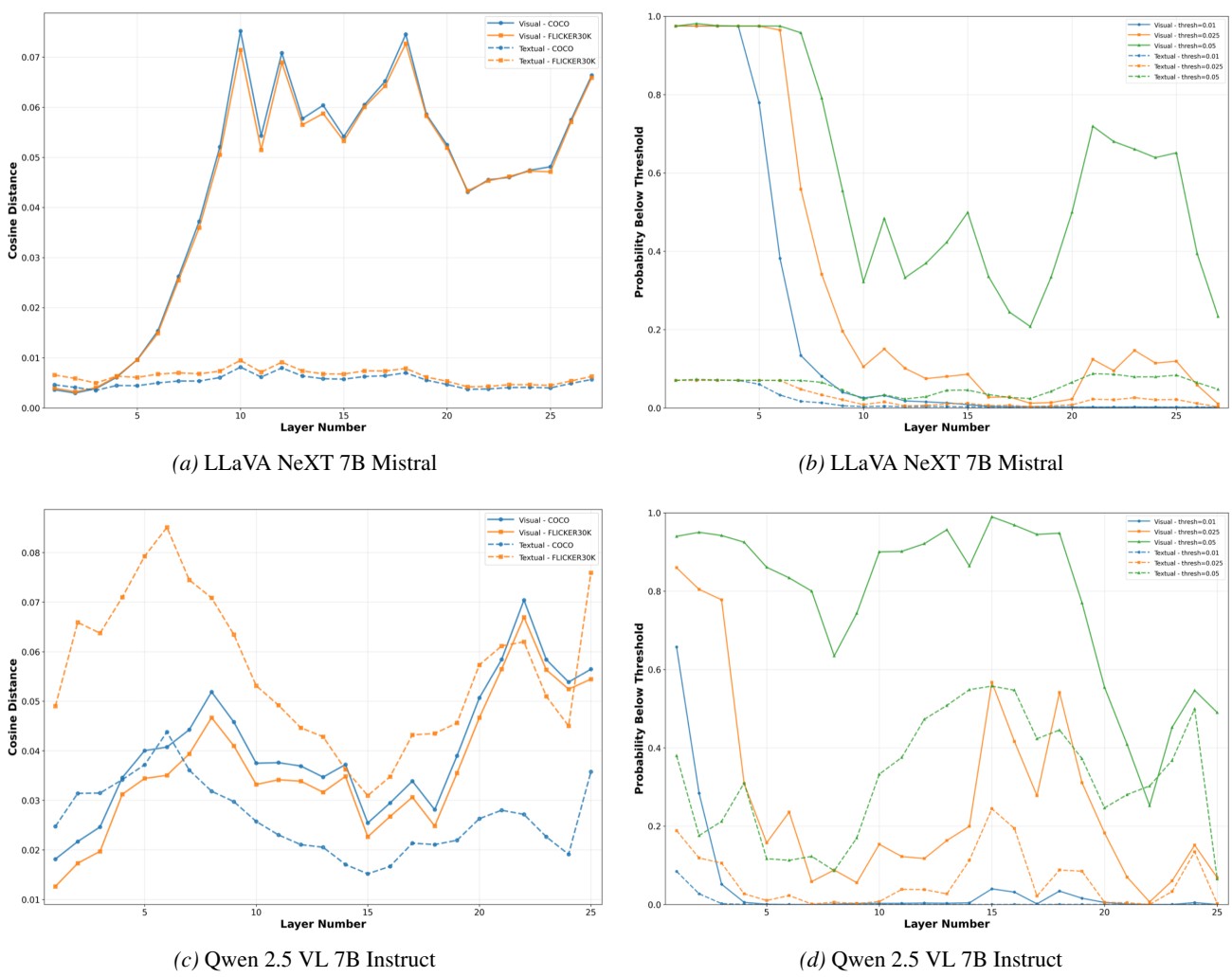

*(a)* LLaVA NeXT 7B Mistral

*(b)* LLaVA NeXT 7B Mistral

*(c)* Qwen 2.5 VL 7B Instruct

*(d)* Qwen 2.5 VL 7B Instruct

*Figure 17.* Empirical Geometric and Proximal Redundancy versus layer. Across the Captioning task (see Table 3) on LLaVA NeXT 7B Mistral and Qwen 2.5 VL Instruct, the early and late layer vision tokens have low adjacent token cosine distances.

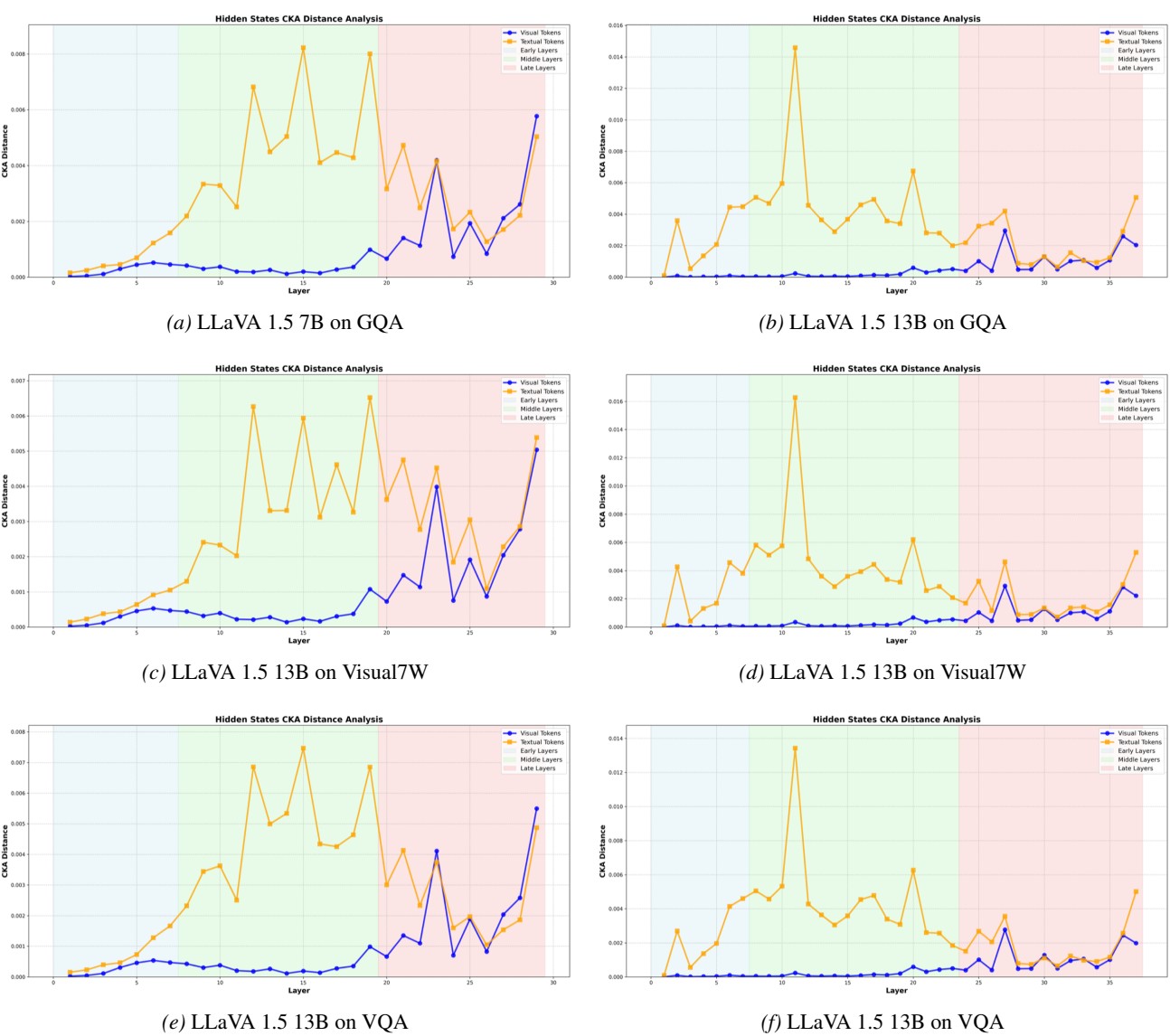

*(a)* LLaVA 1.5 7B on GQA

*(b)* LLaVA 1.5 13B on GQA

*(c)* LLaVA 1.5 13B on Visual7W

*(d)* LLaVA 1.5 13B on Visual7W

*(e)* LLaVA 1.5 13B on VQA

*(f)* LLaVA 1.5 13B on VQA

*Figure 18.* CKA distance (Kornblith et al., 2019) on LLaVA 1.5 architectures. The early vision tokens seem to have low adjacent token CKA distances.

## D. Practical Implications of Section 4.2

Despite the success of our redundancy framework to indicate when early exit and late entry are viable, using one's entire dataset to recreate Section 4.2 is impractical. Therefore in this section, we explore how using a subset of a dataset can theoretically and experimentally generalize the expected layer skipping insights.

### D.1. Experimental Results

In this experiment, we ran the hidden state adjacent cosine distance experiment over the VQA dataset on LlaVA 1.5 and 1.6. We compare the entire VQA dataset adjacent hidden state cosine distance for textual and visual tokens to a 1k sample subset of the dataset. To extract subsets, we set a random seed and uniformly select random samples from each dataset.

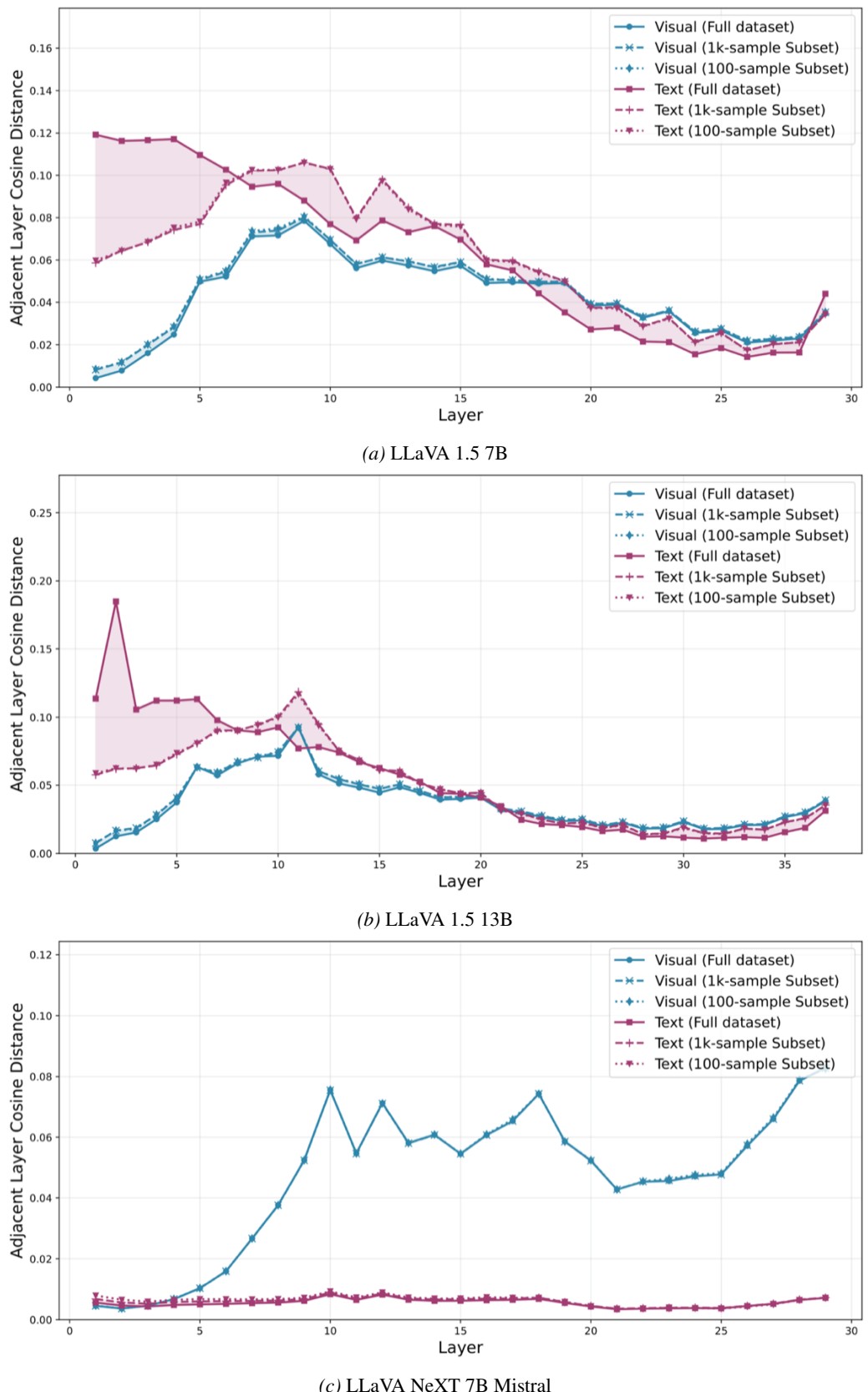

*(a)* LLaVA 1.5 7B

*(b)* LLaVA 1.5 13B

*(c)* LLaVA NeXT 7B Mistral

*Figure 19.* Generalizability of geometric redundancy versus layer for the VQA dataset. The stated subset represents 1k samples from the VQA dataset. LlaVA 1.6 generalizes much more closely than LlaVA 1.5. Note that across these examples, the visual token values are near-identical when averaged over the subset versus the entire dataset.

# E. Further results from Section 4.3

In this section, we include plots of the skipping experiments for LLaVA 1.5 7B/13B and LLaVA NeXT 7B Mistral on all datasets.

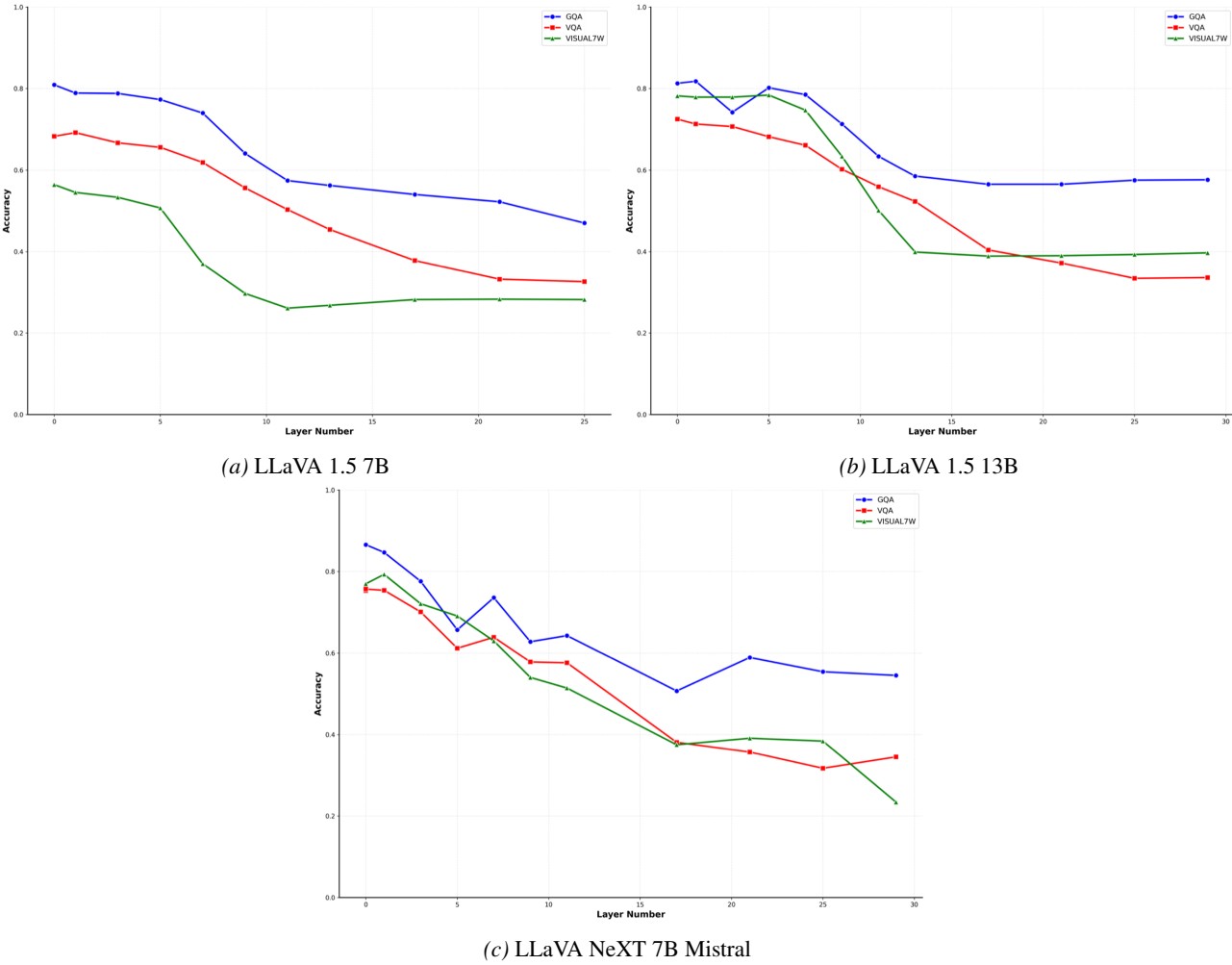

*(a)* LLaVA 1.5 7B                    *(b)* LLaVA 1.5 13B

*(c)* LLaVA NeXT 7B Mistral

*Figure 20.* Skipping model accuracy versus layer. Run across all of the General VQA tasks (see Table 3) on the LlaVA models. The sharpest decrease in the early-middle layers.

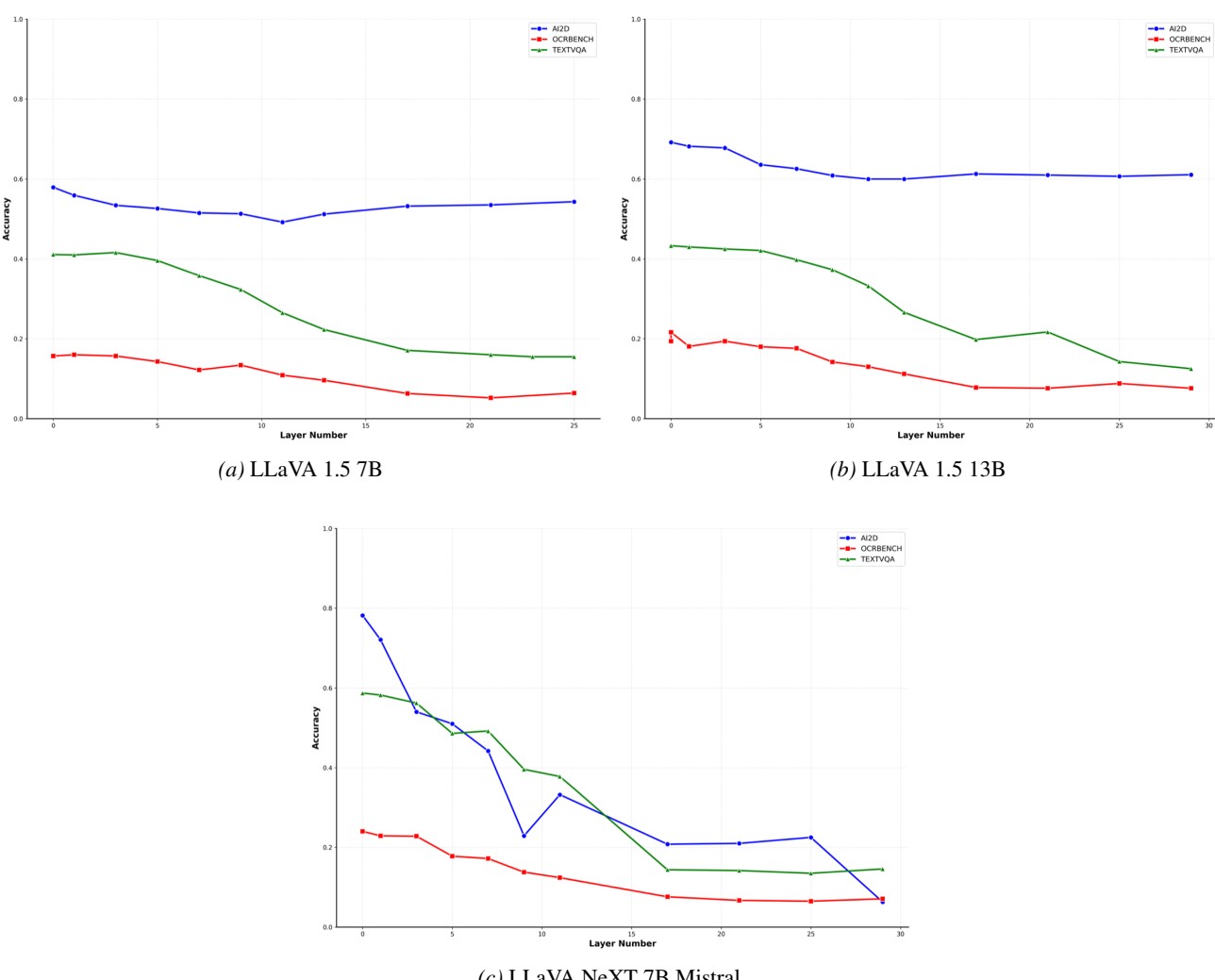

*(a)* LLaVA 1.5 7B

*(b)* LLaVA 1.5 13B

*(c)* LLaVA NeXT 7B Mistral

*Figure 21.* Skipping model accuracy versus layer. Run across all of the Text/Doc VQA tasks (see Table 3) on the LLaVA models. The sharpest decrease in the early-middle layers.

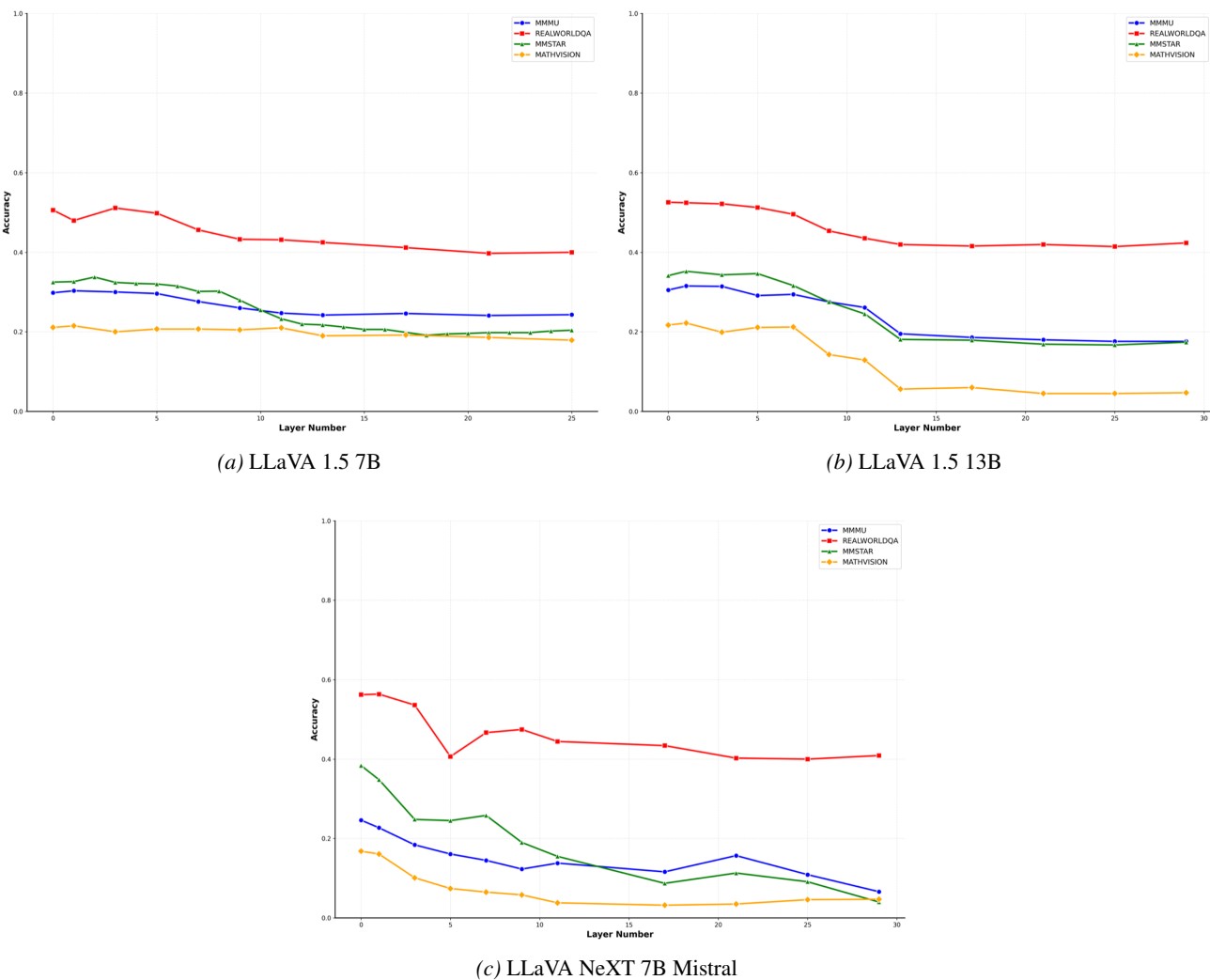

*(a)* LLaVA 1.5 7B

*(b)* LLaVA 1.5 13B

*(c)* LLaVA NeXT 7B Mistral

*Figure 22.* Skipping model accuracy versus layer. Run across all of the Multi-modal reasoning tasks (see Table 3) on the LLaVA models. The sharpest decrease in the early-middle layers.

