# OpenReview forum: "Skip-It? Theoretical Conditions for Layer Skipping in Vision–Language Models"
_ICML.cc/2026/Conference — ICML 2026 regular_

### Official Review · Reviewer_y8Xw · 2026-03-06

**Soundness:** 2
**Presentation:** 3
**Significance:** 3
**Originality:** 2
**Overall Recommendation:** 4
**Confidence:** 3

**Summary:**

Layer skipping techniques for efficiency without sacrificing performance. The authors show experimentally verifiable and interpretable notion for redundancy. The theoretical claim is that geometric/proximal redundancy can imply functional/informational redundancy under assumptions such as Lipschitzness, boundedness, and Markov structure.

**Compliance With Llm Reviewing Policy:**

Affirmed.

**Key Questions For Authors:**

1. How sensitive are the theoretical conclusions to violations of the Lipschitz and Markov assumptions in practice? Did you measure how often these assumptions are violated on the evaluated models?
2. Can the redundancy metrics be used online at inference time for per-sample adaptive skipping, or are they mainly intended as offline diagnostics for choosing a fixed skipping schedule?
3. The paper argues that a small unlabeled subset is sufficient to estimate redundancy. How small is “small” in practice, and how stable are the chosen skip layers across datasets with different image-text distributions?

**Limitations:**

Not needed

**Strengths And Weaknesses:**

Strength
1. The paper gives a taxonomy of redundancy and connects representation closeness to downstream usefulness. That makes the work more foundational than a purely empirical speedup paper.
2. Theoretical claims are aligned with empirical results. Across LLaVA 1.5/1.6 variants and several task families, the paper finds that early vision tokens are often highly redundant, later layers often show low cross-modal attention, and performance drops when skipping is applied outside those regions.
3.
Weakness
1. The paper explicitly finds that captioning behaves differently from VQA/reasoning, with higher later-layer VAR and more task dependence for Late Entry. Informative, but it also weakens any impression of a universal recipe.
2. Theory relies on strong assumptions. The key implications depend on unit-norm representations, Lipschitz conditional expectations, bounded targets, and Markov assumptions. Reasonable from theory perspective but not clear to me how generalized the assumptions are across different VLMs.
3. Your results suggest captioning differs substantially from VQA/reasoning, especially in later-layer attention. What properties of generative decoding make captioning less amenable to early exit?

---

> ### Author Rebuttal · Authors · 2026-03-31
>
> We thank the reviewer for their comments and suggestions. Below, we hope to address their comments.
>
> ```
> Theory relies on strong assumptions...
> ```
>
> We thank the reviewer for their insightful comment on the strength of our assumptions. We address these points as follows:
> - **Unit norm representations**: This assumption is common in Transformer theory (see [4] and [5]). This can further be justified by using the fact that representations lie on a hypersphere ([6]) to then normalize the representations.
> - **Lipschitz**: Similar Lipschitz assumptions are common in this work (see [7]). To supplement this, we have provided estimations of the Lipschitz constants for LLaVA 1.5 and have added estimations for LLaVA NeXT and Qwen 2.5 VL.
> - **Bounded targets**: This assumption is met if we consider the cases of the target being the final layer in the VLM or a classification task. These are the motivating tasks for the theory.
> - **Markov**: If $X_{\ell-1}$ and $X_\ell$ are the representations of layers $\ell-1$ and $\ell$ respectively and $Z$ is the final layer output then $X_{\ell-1}$---$X_\ell$---$Z$ forms a Markov Chain. This is the canonical example explored in our paper.
>
> ```
> The paper explicitly finds that captioning behaves differently from VQA/reasoning...
> ```
>
> We appreciate this observation, but clarify that task-specific differences do not weaken our framework’s universality; instead, they highlight the limitations of current skipping methods. While prior works [1, 2] focus solely on VQA, our work demonstrates that redundancy profiles shift by task, providing a more principled foundation for VLM efficiency. Our contribution is a robust diagnostic framework capable of identifying these nuances, which we believe is more theoretically sound than prescribing a static, potentially sub-optimal skipping policy.
>
> ```
> ... What properties of generative decoding make captioning less amenable to early exit?
> ```
>
> The performance gap in early-exit tasks stems from functional requirements: generative tasks (captioning) are more demanding than discriminative ones (VQA). Pulling from principles in JEPA [3], captioning requires high-fidelity representations to ensure grounded, complex descriptions. Conversely, VQA allows early "collapse" of high-dimensional input into narrow, task-specific features. As an example, answering a specific question about an object’s color requires only a narrow scope of information, making the subsequent layers functionally redundant and thus more viable to an early exit.
>
> In our framework, the "task variable" for captioning represents a comprehensive scene synthesis, while for VQA, it is a targeted semantic feature extracted much earlier in the forward pass.
>
> ```
> How sensitive are the theoretical conclusions to violations of... assumptions...?...
>
> Can the redundancy metrics be used online...?
>
> The paper argues that a small unlabeled subset is sufficient...How small is “small” ...?
> ```
>
> Our theoretical results on functional redundancy are reliant on the Lipschitz assumptions as they become vacuous if violated. As stated in our response to the first question, to help provide evidence towards the assumptions holding in practice, estimates of the Lipschitz constants across layers of some models have been provided. The viability of the Markov assumption was also argued in our response to the first question.
>
> No, redundancy metrics aren't suitable for per-sample adaptive skipping because the decision requires computing the forward pass first. Instead, one can track a running average of geometric and proximal redundancy; once it is sufficiently good---requiring few samples (see below)---this average can guide skipping decisions for subsequent inputs.
>
> In our experiments, textual token redundancy was estimated accurately with 100 samples, while visual tokens required approximately 1k. Given these small sample requirements for offline estimation, online calculation is practically unnecessary.
>
> References
>
> [1] Jiang, et al., "Devil in the Middle Layers: Interpreting, Detecting and Mitigating Object Hallucinations via Attention Lens," CVPR 2025.
>
> [2] Kaduri, et al., "What’s in the image? a deep-dive into the vision of vision-language models," CVPR 2025.
>
> [3] Assran, et al., "Self-Supervised Learning from Images with a Joint-Embedding Predictive Architecture," CVPR 2023.
>
> [4] Karagodin, et al., "Normalization in Attention Dynamics," NeurIPS 2025
>
> [5] Rigollet, "The Mean-Field Dynamics of Transformers," ICM2026
>
> [6] Brody, et al., "On the Expressivity Role of LayerNorm in Transformers’ Attention, " ACL 2023
>
> [7] Zeng, et al., "Skip-Vision: Efficient and Scalable Acceleration of Vision-Language Models via Adaptive Token Skipping," ICCV 2025.

---

> > ### Author Rebuttal · Reviewer_y8Xw · 2026-04-01
> >
> > The rebuttal answers my questions. I will keep the same score 4 - weak accept.

---

> > > ### Author Response · Authors · 2026-04-05
> > >
> > > We would like to thank the reviewer for confirming our rebuttal addressed their concerns. We wanted to also highlight the additional experiments (in rebuttal response to reviewer Ax1X), specifically our hypothesis linking the captioning improvement to spurious attention reduction and shorter generation length and related experimental results. We believe that this additional perspective and experiment further strengthens our rebuttal for the reviewer, and if there are no lingering concerns, we hope the reviewer can consider increasing their score.

---

### Official Review · Reviewer_Ax1X · 2026-03-10

**Soundness:** 2
**Presentation:** 3
**Significance:** 3
**Originality:** 3
**Overall Recommendation:** 5
**Confidence:** 4

**Summary:**

This paper proposes a unified learning and information theoretic framework for characterizing redundancy in vision language models and deriving principled conditions under which layer skipping can be applied without sacrificing downstream performance. The authors define four nested redundancy notions, namely geometric, proximal, functional, and informational, and prove a hierarchy of implications connecting the computationally accessible cosine distance metric to operationally meaningful guarantees on estimator stability and conditional entropy. Layer skipping is formalized into two canonical paradigms, Late Entry and Early Exit of visual tokens, with a formal error bound derived for the former. Empirical validation across LLaVA 1.5 7B/13B, LLaVA NeXT, DeepSeek VL, and Qwen 2.5 VL confirms that early and late layers consistently exhibit high visual token redundancy, and that the proposed conditions correlate with observed performance degradation under skipping.

**Compliance With Llm Reviewing Policy:**

Affirmed.

**Final Justification:**

The reviewers have been very response throughout the discussion period, and have answered my questions, as well as provided substantial additional experiments addressing my concerns. Therefore, I raise my score and recommend Accept.

**Key Questions For Authors:**

**Q1.** The Markov assumption $X_{\ell-1} \text{ --- } X_\ell \text{ --- } Z$ required for Theorem 5 is claimed to follow from residual connections, but this condition requires $X_\ell$ to be a sufficient statistic for the task variable $Z$ relative to $X_{\ell-1}$, which is not implied by the residual structure alone. Can you provide a formal justification or an empirical diagnostic, for example an estimate of $I(Z; X_{\ell-1} \mid X_\ell)$ across layers, that validates or bounds the violation of this assumption?

**Q2.** No formal theorem is provided for Early Exit viability, yet Table 2 constitutes half the empirical contribution. The VAR condition is argued only intuitively. Can you provide a theorem showing that bounded cross modal attention weights imply bounded output error when visual tokens are dropped after layer $\ell$, or explicitly scope the paper's formal claims to Late Entry and reframe Early Exit as an empirically motivated heuristic?

**Q3.** Table 1 shows that Late Entry at layer 4 improves CIDEr on captioning tasks rather than merely preserving it, which the framework does not predict. This pattern is consistent across LLaVA 1.5 7B and 13B. Can you provide a theoretical or empirical explanation for this result, and clarify whether the framework's scope should be qualified for generative versus discriminative tasks?

**Limitations:**

The Impact Statement briefly discusses reduced compute and edge computing applications but omits several substantive limitations. The task and model dependency of the redundancy conditions is framed as a finding rather than a limitation, obscuring the need for per task calibration. The non-operational nature of the theoretical bounds in practice is not acknowledged.

**Strengths And Weaknesses:**

**Soundness** The implication hierarchy connecting proximal and geometric redundancy to functional and informational guarantees is internally coherent, and the MMSE optimality argument in Theorem 1 is technically clean. Theorem 6 is a genuine contribution, making explicit how vision token drift, cross modal dependency, and the Lipschitz properties of the network's tail jointly bound Late Entry output error. However, the Lipschitz constants required by Theorems 1 and 2 are not meaningfully estimated, so the functional redundancy bounds are non-operational in practice. The Markov assumption $X_{\ell-1} \text{ --- } X_\ell \text{ --- } Z$ underpinning Theorem 5 is asserted on the basis of residual connections but not proven or empirically validated. Early Exit receives no formal theorem despite constituting half the empirical contribution, with viability argued only intuitively via VAR. Finally, Table 1 shows that Late Entry systematically improves captioning metrics rather than merely preserving them, a result the framework does not predict and does not explain.

**Presentation** The paper is clearly organized and the redundancy hierarchy is introduced with useful pedagogical structure. The gap between the theoretical results and the empirical measurements is, however, never bridged: no result quantitatively connects the proposed bounds to observed performance gaps, making it impossible to assess whether the theory is tight or merely directionally correct. The cosine distance threshold of 0.05 used throughout Section 4.2.3 is acknowledged as arbitrary but receives no sensitivity analysis or principled selection procedure. Reproducibility is also limited: key experimental details such as whether hidden states are extracted post residual addition or post layer norm, how vision and text tokens are separated across model families, and the sampling procedure for the subset experiments in Appendix D are not specified.

**Significance** The unification of existing methods including AdaSkip, VTW, DeepInsert, SAINT, FlexiDepth, and Skip Vision under a single theoretical lens is genuinely useful to the community, and the label free applicability established by Theorem 7 has direct practical value for deployment without annotation. However, despite framing its contribution around inference efficiency, the paper reports no FLOPs, wall clock latency, or memory measurements anywhere in the submission. The Impact Statement explicitly claims reduced inference time and lower energy consumption, but no efficiency versus accuracy tradeoff is quantified. The finding that redundancy conditions are model and task dependent, most sharply on captioning versus discriminative VQA, means the framework requires per task calibration, which narrows its advantage over existing empirical approaches.

**Originality** The formal justification for cosine distance as a redundancy proxy via Lipschitz arguments, and the connection to Partial Information Decomposition established in Appendix A.3 via Lemma 5, are original contributions relative to prior work that uses these ideas heuristically or does not connect them at all. The core conceptual premise, that layers with similar adjacent representations are safe to skip, is the implicit motivation behind ST$^3$, FastV, and related methods; the paper's novelty lies in formalization rather than conceptual discovery. The formal machinery draws heavily on existing tools from Duchi and Wainwright 2013 and Braun and Pokutta 2015, and this should be acknowledged more explicitly rather than presented as a seamless part of the framework's own development.

---

> ### Author Rebuttal · Authors · 2026-03-31
>
> We thank the reviewer for their comments and constructive feedback. We address their concerns below.
>
> ```
> Q1. The Markov assumption...
> ```
>
> For VLMs, we interpret the task variable $Z$ as representing the final output. Given the residual connection $X_\ell = X_{\ell-1} + \mathcal{F}(X_{\ell-1})$, the Markov property holds: $P(Z\mid X_\ell, X_{\ell-1}) = P(Z\mid X_{\ell})$. This framing aligns directly with our experimental setup.
>
> ```
> Q2. No formal theorem is provided...
> ```
>
> We thank the reviewer for pointing this out and now provide a formal justification for Early Exit in Lemma 5. Informally, if the differences between adjacent vision representations and the impact of vision on text are small for the final layers, then the impact of early exit is minimal. The proof mimics the Late Entry theorem.
>
> ```
> Q3. Table 1 shows that Late Entry...
> ```
>
> Middle layers typically handle visual processing [1, 2], but we hypothesize that the nature of the task plays a key role here. While VQA often refers to specific parts of the scene, captioning requires describing the whole image. In that sense, this skipping can mitigate spurious attention---where models can fixate on irrelevant correlations [1]---reducing hallucinations to greater degree for captioning (Figures 14 and 15 support this). Additionally, we found that early skipping often lowers average generation length, which could contribute to improving these metrics.
>
> ```
> ...the Lipschitz constants...are not meaningfully estimated...
> ```
>
> ```
> The gap between theory and empirical...
> ```
>
> We provide estimates of the Lipschitz constants for LLaVA 1.5 7B and now add estimates for LLaVA NeXT and Qwen 2.5 VL in Appendix A.4. We acknowledge that the provided bounds are directional by design, as they rely on Lipschitz assumptions, which makes them not very tight. Nevertheless, such assumptions are standard in prior work [3]. Our theory aims to explain why measurable metrics, such as average cosine distance, can serve as viable proxies for more informative measures, such as functional or informational redundancy.
>
> ```
> The cosine distance threshold... receives no sensitivity analysis...
> ```
>
> We now add a sensitivity analysis for $\epsilon \in [0.001,0.1]$. We found minimal differences in which layers should be skipped for $\epsilon \in [0,03,0.06]$. As can be seen in Figure 3, if $\epsilon<0.01$, the framework will not have any redundancy in the late layers, and if $\epsilon>0.08$, all layers will be marked as redundant for vision. We will include this analysis in the appendix.
>
> ```
> ...key experimental details...are not specified.
> ```
>
> Hidden states are extracted post residual addition. LLaVA 1.5 vision tokens are contained in a fixed contiguous block of 576 tokens, so the vision tokens are isolated by finding this block starting from the index of the $<$image$>$ token. LLaVA NeXT, Qwen 2.5 VL, and Deepseek-VL use dynamic token counts, so we search for a contiguous block of image token ids. In the subset procedure, we set a random seed and uniformly select $n$ random samples from each dataset. We also add these details to the appendix.
>
> ```
> ...the paper reports no FLOPs... claims reduced inference time...
> ```
>
> We would like to emphasize that our contribution in this work is providing a unified framework for existing layer-skipping techniques and not a novel algorithm. The framework induces a simple algorithm for skipping, which reduces to existing "late entry" and "early exit" methods. As such, for an in-depth analysis of efficiency improvements, refer to these existing works (e.g., [4] and [5]).
>
> ```
> ...redundancy conditions are model and task dependent...
> ```
>
> We appreciate this observation, but clarify that task-specific differences do not weaken our framework’s universality; instead, they highlight the limitations of current skipping methods. While prior works [1, 2] focus solely on VQA, our work demonstrates that redundancy profiles shift by task, providing a more principled foundation for VLM efficiency. Our contribution is a robust framework capable of identifying these nuances, which we believe is more theoretically sound than prescribing a static, potentially sub-optimal skipping policy.
>
> ```
> The formal machinery draws heavily on existing tools...
> ```
>
> We thank the reviewer and will add clearer acknowledgments to our reliance on the past work.
>
> References
>
> [1] Jiang, et al., "Devil in the Middle Layers: Interpreting, Detecting and Mitigating Object Hallucinations via Attention Lens," CVPR 2025.
>
> [2] Kaduri, et al., "What’s in the image? a deep-dive into the vision of vision-language models," CVPR 2025.
>
> [3] Zeng, et al., "Skip-Vision: Efficient and Scalable Acceleration of Vision-Language Models via Adaptive Token Skipping," ICCV 2025.
>
> [4] Choraria, et al., "DeepInsert: Early Layer Bypass for Efficient and Performant Multimodal Understanding," EACL 2026.
>
> [5] Elhoushi et al., "LayerSkip: Enabling Early Exit Inference and Self-Speculative Decoding," ACL 2024.

---

> > ### Author Rebuttal · Reviewer_Ax1X · 2026-04-03
> >
> > The rebuttal sufficiently addressed my concerns regarding the Markov assumption (Q1) and the missing formal justification for Early Exit in Lemma 5 (Q2). My concern regarding Q3 was partially addressed via a plausible hypothesis linking the captioning improvement to spurious attention reduction and shorter generation length, however insufficent empirical evidence is provided to support it. I maintain my score of 4.

---

> > > ### Author Response · Authors · 2026-04-05
> > >
> > > We ran additional experiments to help empirically justify the claims we made in our initial response. We experimentally computed the average generation length for Captioning versus VQA tasks to support our answer to Q3. For open-generation VQA, the average generation length was 8.24 tokens, compared to 14.87 tokens for captioning. These results are provided in the table:
> > >
> > > Layer | Baseline | 4 | 8 | 12
> > >
> > > VQA | 9.012 | 8.634 | 8.356 | 6.964
> > >
> > > Captioning | 20.90 | 13.82 | 12.74 | 12.03
> > >
> > > Based on the table, as the number of layers skipped increases, the number of generated tokens decreases.
> > >
> > > We also have an experiment results regarding the reviewer's question on inference speed of the layer skipping strategy we used in our work.
> > >
> > > We compute the theoretical FLOPs using Equation 5 in "DeepInsert: Early Layer Bypass for Efficient and Performant Multimodal Understanding," $\text{FLOPs}(N_{\text{DI}}) = N \Big( 8(L_{\text{text}} + L_{\text{mm}}) d_{\text{model}}^2 + 4(L_{\text{text}} + L_{\text{mm}})^2 d_{\text{model}} + 4(L_{\text{text}} + L_{\text{mm}}) d_{\text{model}} d_{\text{ff}}) - N_{\text{DI}} \Big( 8 L_{\text{mm}} d_{\text{model}}^2 + 4(2L_{\text{text}} + L_{\text{mm}}) L_{\text{mm}} d_{\text{model}} + 4 L_{\text{mm}} d_{\text{model}} d_{\text{ff}})$, with LLaVA-7B (N=32, d_model=4096, d_ff=11008, n_head=32, L_mm=576, L_text=256). We provide the latency per sample and TFLOPs at each late entry layer. (we found that late entry has very similar results, for the same number of layers skipped.) We used a single single node on an NVIDIA GH200 Grace Hopper Superchip.
> > >
> > > Early Exit Layer | TFLOPs | Latency (s):
> > >
> > > Baseline | 5.47 | 1.74
> > >
> > > 6 | 4.84 | 1.54
> > >
> > > 10 | 4.42 | 1.28
> > >
> > > 14 | 3.99 | 1.16
> > >
> > > 18 | 3.57 | 0.83
> > >
> > > 22 | 3.15 | 0.64
> > >
> > > 26 | 2.72 | 0.51
> > >
> > > 30 | 2.30 | 0.48
> > >
> > > From the table, is a close match between the theoretically estimated FLOPs of (5) and the empirically measured runtime of the forward pass of the model.
> > >
> > > We hope that these additional experiments adequately address the reviewer's questions.

---

### Official Review · Reviewer_oEnx · 2026-03-14

**Soundness:** 3
**Presentation:** 2
**Significance:** 2
**Originality:** 3
**Overall Recommendation:** 4
**Confidence:** 3

**Summary:**

This paper presents a theoretical framework for understanding when layer skipping in VLMs is safe and effective. It introduces multiple notions of redundancy (proximal, geometric, informational, functional), derives implication relations under regularity assumptions, and unifies existing skipping strategies into Early Exit and Late Entry. The empirical section verifies that early and late visual layers often exhibit redundancy across several VLM backbones and benchmarks.

**Compliance With Llm Reviewing Policy:**

Affirmed.

**Final Justification:**

I have carefully read the author rebuttal and appreciate the detailed clarifications and additional analyses. The responses have addressed my main concerns regarding practicality and completeness. While some limitations remain (e.g., assumption strength and limited competitive evaluation), the paper now presents a more complete and well-supported contribution. Overall, the rebuttal improves my assessment of the work, and I updated my rating to weak accept.

**Key Questions For Authors:**

1. The inference speed / FLOPs measurement is important to evaluate the efficiency of layer-skipping methods, could you provide the inference speed and accuracy of models with the proposed method, comparing with existing layer-skipping works?

**Limitations:**

yes

**Strengths And Weaknesses:**

Strengths:
1. The paper provides a unified lens to interpret many existing layer-skipping methods, which is helpful for the field.
2. The proposed measurable redundancy indicators (e.g., cosine/proximal metrics, attention ratio) can be tested without labels.

Weaknesses:
1. Method novelty is limited on the algorithmic side: the paper mainly explains and unifies existing ideas rather than proposing a clearly stronger skipping algorithm.
2. Evaluation focus is explanatory rather than competitive: limited direct comparisons against strongest recent skipping baselines under matched compute/latency budgets.
3. Assumption-heavy theory: key guarantees rely on Lipschitz/Markov/boundedness assumptions; practical tightness and robustness are not fully characterized.

---

> ### Author Rebuttal · Authors · 2026-03-31
>
> We thank the reviewer for their comments and constructive feedback. We address their concerns below.
>
> ```
> Method novelty is limited on the algorithmic side: the paper mainly explains and unifies existing ideas rather than proposing a clearly stronger skipping algorithm.
> ```
>
> We respectfully disagree with the sentiment of limited novelty. The current literature in VLM efficiency has primarily focused on new layer skipping and pruning methods to speed up inference by removing redundant computation. The success of these methods imply redundancy in VLMs, but such redundancy has not been explored beyond measuring downstream performance on a limited set of benchmarks. In that sense, one of our key contributions is to be able to do this in a model and task agnostic manner with theoretical guarantees.
>
> The framework we propose can be naturally extended into an algorithm, which we now add. For brevity, we request the reviewer to refer to point (1) in response to reviewer Jt85.
>
> ```
> Evaluation focus is explanatory rather than competitive: limited direct comparisons against strongest recent skipping baselines under matched compute/latency budgets.
> ```
>
> We wish to reiterate that our contribution lies in identifying redundancy within VLMs and explaining when layer skipping is possible, rather than a new skipping technique. Unlike existing methods, our work pinpoints not just layers are redundant but also provides the underlying verifiable rationale. By demonstrating how tokens can bypass specific layers using standard Early Exit and Late Entry mechanisms, we establish a framework that future research can leverage to design more informed skipping strategies.
>
> ```
> Assumption-heavy theory: key guarantees rely on Lipschitz/Markov/boundedness assumptions; practical tightness and robustness are not fully characterized.
> ```
>
> We agree with the reviewer that these are heavy assumptions which underlie our theory, however, these assumptions are consistently made in ML theory for Transformers. For example, [1] uses similar Lipschitz assumptions to argue when certain vision tokens can be pruned. Furthermore, commonly in the Information Bottleneck literature (see [2]), the adjacent layer outputs are seen as forming a Markov Chain. In our case, if we consider the setting of layer $\ell-1$, $\ell$, and the final VLM output as is done in our experiments, $X_{\ell-1}$---$X_{\ell}$---$Z$ forms a Markov Chain. This is because the layer representations are taken after residual additions. Furthermore, the boundedness assumption is common in Transformer theory literature and has been argued and used in [3] and [4] for analyzing the mean-field dynamics of Transformers. Additionally, LayerNorm [5] empirically appears to restrict layer representations to a hypersphere, so one can also consider the random variables of interest as the normalized layer representations. To further provide support for our Lipschitz assumptions, we add further experimentation on the approximate Lipschitz constants for LLaVA NeXT 1.6 Mistral and Qwen 2.5 VL 7B to Appendix A.4.
>
> ```
> Key Question: The inference speed / FLOPs measurement is important to evaluate the efficiency of layer-skipping methods, could you provide the inference speed and accuracy of models with the proposed method, comparing with existing layer-skipping works?
> ```
>
> We would like to emphasize that our contribution in this work is in providing a unified framework for existing layer skipping and pruning techniques and not a novel layer-skipping method. The framework can be extended to a simple algorithm for layer skipping decisions, however, it reduces to existing "late entry" and "early exit" skipping strategies. As such, for in-depth analysis of inference speed-ups and FLOPs, we direct the reviewer towards these works (see [6], [7]). Specifically, the DeepInsert [6] late entry method showed a per-token inference time decrease from 100ms to 93ms when injecting at layer 4 in LLaVA 1.5 7B and showed a linear decrease in the number of FLOPs with respect to the injection layer. For the LayerSkip [7] early exit method, they show a per-token inference time decrease from 165ms to 124ms when exiting at layer 18 on Llama 1.5B.
>
> References
>
> [1] Weili Zeng, et. al, "Skip-Vision: Efficient and Scalable Acceleration of Vision-Language Models via Adaptive Token Skipping," ICCV 2025.
>
> [2] Kenji Kawaguchi, et. al, "How Does Information Bottleneck Help Deep Learning?" ICML 2023
>
> [3] Nikita Karagodin, et. al, "Normalization in Attention Dynamics," NeurIPS 2025
>
> [4] Phillipe Rigollet, "The Mean-Field Dynamics of Transformers," ICM2026
>
> [5] Shaked Brody, et. al, "On the Expressivity Role of LayerNorm in Transformers’ Attention," ACL 2023
>
> [6] Moulik Choraria, et. al, "DeepInsert: Early Layer Bypass for Efficient and Performant Multimodal Understanding," EACL 2026.
>
> [7] Mostafa Elhoushi et. al., "LayerSkip: Enabling Early Exit Inference and Self-Speculative Decoding," ACL 2024.

---

> > ### Author Rebuttal · Reviewer_oEnx · 2026-04-05
> >
> > I have carefully read the author rebuttal and appreciate the detailed clarifications and additional analyses. The responses have addressed my main concerns regarding practicality and completeness. While some limitations remain (e.g., assumption strength and limited competitive evaluation), the paper now presents a more complete and well-supported contribution. Overall, the rebuttal improves my assessment of the work, and I will update my rating to weak accept.

---

> > > ### Author Response · Authors · 2026-04-05
> > >
> > > We thank the reviewer for their thoughtful review of our work and subsequent rebuttal. We added additional experiments to help address the reviewer's key question regarding latency in our response to reviewer Ax1X.
> > >
> > > The reviewer felt that our rebuttal improved their assessment of our work and would raise their review to a weak accept (4). We politely request that the reviewer officially raise their score in the system to reflect this new assessment.

---

### Official Review · Reviewer_Jt85 · 2026-03-15

**Soundness:** 4
**Presentation:** 4
**Significance:** 4
**Originality:** 4
**Overall Recommendation:** 4
**Confidence:** 3

**Summary:**

This paper proposes a unified theoretical framework for characterizing redundancy in Vision-Language Models (VLMs), with the goal of providing principled conditions under which layer skipping can be applied without significant performance degradation. The authors define four notions of redundancy and prove formal relationships among them.  They further provide a generalization bound showing that redundancy can be estimated from a small subset of unlabeled data.

**Compliance With Llm Reviewing Policy:**

Affirmed.

**Key Questions For Authors:**

see above.

**Limitations:**

yes

**Strengths And Weaknesses:**

Strengths：
1. The paper defines four distinct notions of redundancy and rigorously proves their inter-relationships.
2. The paper explicitly tests whether the theoretical redundancy conditions predict actual performance degradation when layers are skipped.

Weaknesses:
1. The paper provides conditions for when layer skipping is safe but does not propose an algorithm to automatically select which layers to skip based on these conditions.
2. The paper does not provide theoretical justification for why certain tasks should exhibit different redundancy profiles, nor does it offer task-adaptive thresholds.

---

> ### Author Rebuttal · Authors · 2026-03-31
>
> We thank the reviewer for appreciating the theoretical rigor and depth of empirical validation of our work, as well as their constructive feedback. We address their concerns below.
>
> ```
> The paper provides conditions for when layer skipping is safe but does not propose an algorithm to automatically select which layers to skip based on these conditions.
> ```
>
> While our results do imply a natural algorithmic extension for practical scenarios, we agree with the reviewer's observation that this should be explicitly stated. To that end, we update the manuscript to provide the requisite algorithm to automate the choosing of the layers for skipping based on a given data subset and our theoretical conditions. The algorithm is as follows:
>
> - Given a subset of data D, and threshold values $\epsilon$ (geometric redundancy distance threshold), $t$ (proximal redundancy  distance threshold), $\alpha$ (proximal redundancy probability threshold). In practice, these parameters should be selected based on the desired trade-off of performance degradation and efficiency gain. They can be validated by skipping their suggested layers and checking for model degradation.
>
> - For each example x in D, run a forward pass and collect hidden states.
>
> - For each layer, compute the average $t$-proximal, geometric redundancy, and visual attention ratio over D.
>
> - (For late entry) Search for the last layer from the beginning such that all previous layers' average $t$-proximal and geometric redundancy meet redundancy thresholds.
>
> - (For early exit) Search for the first layer from the end such that all future layers' average $t$-proximal, geometric redundancy, and VAR meet thresholds.
>
> We have also added this as a standalone script in our code to enable ease of deployment. We will update the manuscript when given a chance to do so.
>
> ```
> The paper does not provide theoretical justification for why certain tasks should exhibit different redundancy profiles, nor does it offer task-adaptive thresholds.
> ```
>
> Regarding task specific dependency, we want to stress that the fact that layer skipping is not just model but also task dependent is an important contribution of our work. Previous works [1, 2] advocate for skipping techniques which are evaluated exclusively on VQA datasets. The task specific dependency challenges the generalizability of these approaches. On the point of a theoretical justification, we want to clarify the scope of our contribution, which is to principally determine when layer skipping is feasible. The question of "why" it is feasible in the first place (let alone the task specific behavior) is still an open question in the community, and will likely involve a careful mechanistic analysis of the cross-modal circuits of these models. In that sense, captioning requires even further work to consolidate the behavior across multiple forward passes for each newly generated token.
>
> Nonetheless, we provide a hypothesis based on current literature. Previous works on VLMs [3, 4] demonstrated that visual information processing is primarily done in the middle model layers. We hypothesize that early layers first process textual tokens to determine the specific information required from subsequent visual tokens. This processing demand varies by task: discriminative VQA queries are typically more complex and require more intensive early-layer analysis than simpler captioning prompts. Formally, this distinction is captured by the task variable $Z$ in our functional redundancy framework. While a VQA output token $Z$ may represent abstract information not directly copied from visual inputs, the captioning task requires $Z$ to be extracted directly from fine-grained visual details to ensure model faithfulness.
>
> With regards to task specific thresholds, we point out that in our experiments, we used the same threshold hyper-parameters for VQA and captioning tasks. Based on tables 1 and 2, setting $\epsilon=0.05$ for a geometric redundancy threshold and $\alpha=0.97$ for a $0.05$-proximal redundancy threshold correctly predicts when layers can be skipped with minimal degradation for VQA. For Captioning, these thresholds should be set slightly less strict: $\epsilon=0.07$ and $\alpha=0.95$. Our algorithm above is able to automatically identify the viable layers across different models and tasks, and a set of hyper-parameters can be easily identified and verified.
>
> References
>
> [1] Zhuomin He, et. al, "AdaSkip: Adaptive Sublayer Skipping for Accelerating Long-Context LLM Inference," AAAI 2025.
>
> [2] Weili Zeng, et. al, "Skip-Vision: Efficient and Scalable Acceleration of Vision-Language Models via Adaptive Token Skipping," ICCV 2025.
>
> [3] Zhangqi Jiang, et. al, "Devil in the Middle Layers: Interpreting, Detecting and Mitigating Object Hallucinations via Attention Lens," CVPR 2025.
>
> [4] Omri Kaduri, et. al, "What’s in the image? a deep-dive into the vision of vision-language models," CVPR 2025.

---

> > ### Author Rebuttal · Reviewer_Jt85 · 2026-04-03
> >
> > My concerns have been adequately addressed.

---

> > > ### Author Response · Authors · 2026-04-04
> > >
> > > We are pleased to hear that our responses and the additions to the manuscript, specifically the formal algorithm for layer selection and the clarified discussion on task-adaptive thresholds, have addressed your concerns.
> > >
> > > As you noted in your acknowledgement that the concerns are fully resolved, would you consider raising your official score to reflect this updated assessment?
> > >
> > > We believe that with the inclusion of the practical algorithm and the additional theoretical justifications provided during this discussion, the manuscript is now significantly strengthened and provides a more complete contribution to the study of redundancy in VLMs.
> > >
> > > Additionally, we provided new experiments related to why different task types may exhibit different redundancy profiles (See response to reviewer Ax1X's acknowledgement).
> > >
> > > Thank you again for your guidance in improving this work.

---

### Decision · Program_Chairs · 2026-04-30

**Decision:**

Accept (regular)

**Comment:**

The paper looks into how to quantify redundancy in VLMs, both to prune tokens and to skip layers. Here, a layer is redundant if it acts as an identity function, whereas tokens are redundant if their representations encode duplicate information. The authors establish the theoretical relationships that link several definitions of redundancy into a formal hierarchy. They then validate their analysis on models such as LLaVA 1.5 7B/13B, LLaVA NeXT, DeepSeek VL, and Qwen 2.5 VL.

The paper is well-written and the methodology is sound. Given the unanimous positive reviews, I recommend acceptance.

While I encourage the authors to address all of the comments raised by the reviewers, some concerns that they should pay special attention to include:

* Sensitivity analysis should be included for the cosine distance threshold.

* The paper should provide more details to facilitate reproducibility (e.g. how the hidden features are extracted exactly).

* Please make sure to update the paper as promised in the rebuttal (e.g. by including a clear description of the algorithm).